# A new class of capsid-targeting inhibitors that specifically block HIV-1 nuclear import

Aude Boulay[1], Emmanuel Quevarec[1], Isabelle Malet[2], Giuseppe Nicastro [3], Célia Chamontin [1], Suzon Perrin[1], Corinne Henriquet[4], Martine Pugnière[4], Valérie Courgnaud[5], Mickaël Blaise [1], Anne-Geneviève Marcelin[2], Ian A Taylor [3], Laurent Chaloin [1] & Nathalie J Arhel [1]✉

## Abstract

HIV-1 capsids cross nuclear pore complexes (NPCs) by engaging with the nuclear import machinery. To identify compounds that inhibit HIV-1 nuclear import, we screened drugs in silico on a three-dimensional model of a CA hexamer bound by Transportin-1 (TRN-1). Among hits, compound H27 inhibited HIV-1 with a low micromolar $IC_{50}$. Unlike other CA-targeting compounds, H27 did not alter CA assembly or disassembly, inhibited nuclear import specifically, and retained antiviral activity against PF74- and Lenacapavir-resistant mutants. The differential sensitivity of divergent primate lentiviral capsids, capsid stability and H27 escape mutants, together with structural analyses, suggest that H27 makes multiple low affinity contacts with assembled capsid. Interaction experiments indicate that H27 may act by preventing CA from engaging with components of the NPC machinery such as TRN-1. H27 exhibited good metabolic stability in vivo and was efficient against different subtypes and circulating recombinant forms from treatment-naïve patients as well as strains resistant to the four main classes of antiretroviral drugs. This work identifies compounds that demonstrate a novel mechanism of action by specifically blocking HIV-1 nuclear import.

**Keywords** HIV-1; Capsid; Nuclear Import; Drug Discovery; Antiretroviral Therapy
**Subject Categories** Microbiology, Virology & Host Pathogen Interaction; Pharmacology & Drug Discovery

## Introduction

Retroviruses replicate in the nucleus of their target cells by integrating their genome into the host chromatin. While most retroviruses require a nuclear envelope breakdown to access the nucleus, lentiviruses such as HIV-1 have evolved the capacity to pass through nuclear pore complexes (NPCs) by active transport (Bukrinsky et al, 1992). This remarkable property has been harnessed to develop lentiviral vectors that are used in the laboratory and the clinic to stably transfer genes in non-dividing cells and the underlying mechanisms have been extensively studied over the past three decades.

Following recent reports demonstrating that viral complexes still contain intact capsid shells at the NPC and in the nucleus (Arhel et al, 2007; Burdick et al, 2020; Dharan et al, 2020; Li et al, 2021; Muller et al, 2021a; Selyutina et al, 2020; Zila et al, 2021), our current understanding is that the HIV-1 genome does not enter the nucleus as a naked DNA but is carried across by its viral capsid. The HIV-1 core is an assembly of ~1500 p24 capsid monomers (hereafter called CA), into 200–250 hexamers and 12 pentamers, forming a fullerene cone (Wilbourne and Zhang, 2021). The CA protein, which is the only protein component of the capsid shell, consists of two distinct domains connected by a flexible linker: a N-terminal domain (NTD) composed of one β-hairpin and 7 α-helices (1 to 7), which faces the outer surface of the core towards the cellular environment, and a C-terminal domain (CTD) comprised of α-helices 8 to 11, which faces the inside of the core. Capsid stability, which is essential for infectivity (Forshey et al, 2002), is controlled by several inter- and intra-subunit interactions and numerous host cell factors (Jang and Engelman, 2023; Wilbourne and Zhang, 2021).

HIV-1 nuclear entry is likely mediated by multiple cumulative and successive interactions between viral complexes and host factors. The viral determinants of nuclear import are generally attributed to the HIV-1 CA and integrase, both of which possess karyophilic elements that facilitate the passage of pre-integration complexes (PICs) across NPCs (Bouyac-Bertoia et al, 2001; Fernandez et al, 2019; Gallay et al, 1997; Yamashita and Emerman, 2004). Despite their size, it is thought that HIV-1 capsids successfully overcome the diffusional barrier of NPCs in a number of ways (Guedan et al, 2021; Taylor and Fassati, 2023; Yamashita and Engelman, 2017), for instance by interacting with nuclear transport receptors (NTRs, also called karyopherins) such as Transportin-1 and -3 (Brass et al, 2008; Christ et al, 2008; Fernandez et al, 2019; Krishnan et al, 2010; Valle-Casuso et al, 2012),

[1]Institut de Recherche en Infectiologie de Montpellier (IRIM), University of Montpellier, CNRS 9004, 34293 Montpellier, France. [2]Department of Virology, INSERM, Sorbonne University, AP-HP, Pitié-Salpêtrière Hospital, Paris, France. [3]Macromolecular Structure Laboratory, The Francis Crick Institute, 1 Midland Road, London NW1 1AT, UK. [4]Institut de Recherche en Cancérologie de Montpellier, INSERM, University of Montpellier, Institut Régional du Cancer, Montpellier, France. [5]RNA viruses and host factors, Institut de Génétique Moléculaire de Montpellier, University of Montpellier, CNRS-UMR 5535, 1919 Route de Mende, Montpellier 34293, Cedex 5, France. ✉E-mail: nathalie.arhel@irim.cnrs.fr

by directly engaging with FG-nucleoporins such as Nup358, Nup153 and Nup98 (Di Nunzio et al, 2012; Di Nunzio et al, 2013; Dickson et al, 2024; Fu et al, 2024; Matreyek et al, 2013; Schaller et al, 2011; Shen et al, 2023a; Shen et al, 2023b; Xue et al, 2023), and by recruiting host cell factors that are critical for capsid stability, such as CypA and IP6, or early post-nuclear entry events, such as the cellular protein cleavage and polyadenylation specificity factor 6 (CPSF6) (Bhargava et al, 2018; Engelman, 2021; Hilditch and Towers, 2014; Taylor and Fassati, 2023).

There are currently over 20 antiretroviral drugs that are approved for the treatment of persons living with HIV, including nucleoside (NRTIs) and non-nucleoside (NNRTIs) reverse transcription inhibitors, protease inhibitors (PIs) and integrase strand transfer inhibitors (INSTIs). When used in combination, they are highly effective and provide patients with a life expectancy similar to individuals without HIV, and also substantially reduce virus transmission (Ghosn et al, 2018). More recently, the HIV capsid has emerged as an attractive therapeutic target for novel antiretrovirals due to its importance in the early and late stages of the viral life cycle (McFadden et al, 2021). Small molecule inhibitors that target a common site for binding of HIV-1 capsid by CPSF6 and Nup153 have been developed (Engelman, 2021). The compound PF-3450074 (PF74) inhibits HIV-1 replication in the low micromolar range (Shi et al, 2011), whereas GS-CA1 (Yant et al, 2019) and GS-6207 (Lenacapavir, LEN) (Bester et al, 2020; Link et al, 2020) are potent low nanomolar to picomolar inhibitors. Moreover, LEN has undergone clinical trials as a long-acting antiretroviral (Gupta et al, 2023; Segal-Maurer et al, 2022) and is now approved for clinical use by the US FDA.

While antiretroviral therapy (ART) suppresses viral loads to below detection levels, it does not reverse the integration of the viral DNA within the host cell genome. This permanent potential for virion production fuels episodic and predominantly undetectable viral replication throughout the lifetime of the patient (Martinez-Picado and Deeks, 2016), which favours innate immune activation and maintains the virus reservoir (Bachmann et al, 2019; Castro-Gonzalez et al, 2018).

Despite ART, HIV-1 persists in the nucleus of infected cells in the form of integrated and unintegrated provirus, both of which can produce viral transcripts and some viral proteins that fuel chronic and deleterious immune activation (Boulay et al, 2023; Sloan and Wainberg, 2011). None of the major classes of antiretroviral drugs, namely those that target reverse transcription, integration and maturation, prevent HIV from reaching the nuclear compartment, and a specific inhibitor of HIV nuclear entry has yet to be identified.

We hypothesised that compounds that prevent HIV-1 capsids from engaging with the nuclear import machinery of their target cells could have antiviral properties, by disrupting uncoating, nuclear import, or both. Using a 3D model of HIV-1 CA hexamers bound by the karyopherin TRN-1 based on our previous mutant studies (Fernandez et al, 2019), we searched chemical databases for compounds that might bind at this interface. The screen led to the identification of one hit compound, called H27, which inhibited HIV-1 infection with low micromolar potency and negligible toxicity, including in primary human lymphocytes. Further work showed that H27 and five related and commercially available analogues inhibited the nuclear import of HIV-1 specifically, without impacting other steps of the viral life cycle, such as reverse transcription or particle production. Our work, therefore, introduces an original series of HIV-1 antivirals that specifically inhibit HIV-1 nuclear import by targeting the HIV-1 CA.

## Results

### Virtual screening at the HIV-1 CA hexamer–TRN-1 interface identifies H27 as an antiviral molecule

To discover potential HIV-1 nuclear import inhibitors, we performed in silico screening on the CA hexamer–TRN-1 complex, using a 3D model based on previous mutagenesis studies (Fernandez et al, 2019). The 3D coordinates of the complex were first prepared by applying short molecular dynamics (MD) simulations in explicit water to evaluate its binding stability, followed by a simulation at constant temperature (310 K) and to finally end up with a relaxed complex by minimising its potential energy (see method section for details). From this step, a virtual screening campaign was started on the energy-minimised complex and by targeting the main interface between CA and TRN-1 proteins (defined as a sphere of 10 Å radius around Thr521 from TRN-1 which is positionned in front of the CA PR loop). Large chemical databases, such as ChEMBL25 and MolPort, were pre-filtered using physicochemical criteria typical of protein/protein interfaces (iPPIs) (Basse et al, 2013; Labbe et al, 2016) to select molecules with strong iPPI potential and virtual screening was performed using the GOLD programme (see Fig. EV1 for an overview of the process). The resulting hits were ranked according to their docking score, which reflects the theoretical binding affinity for the target. The 40 best-ranked hits were then selected according to their chemical diversity and availability from commercial resellers, and purchased for further experimental evaluation (Dataset EV1). A quick overview of the chemical structure of the best hits indicates that most of the compounds are composed of several aromatic rings and, in many cases, a sulfonyl, sulfonate or sulfonamide group (hits **1, 4, 6, 23, 27, 30, 31, 34** and **35**).

Compounds **1** to **40** (Appendix Fig. S1) were initially screened using an HIV-1 derived vector LV-CMV-NLuc in HeLa-R5 cells stably expressing Luciferase to normalise for cellular input and toxicity. Compound **27** (4-[(5-tert-butyl-2-methylphenyl)sulfonylamino]-*N*-[4-methoxy-3-[(4-methoxyphenyl)sulfamoyl]phenyl] benzamide, $C_{32}H_{35}N_3O_7S_2$, MW = 637.77, hereafter H27) was the only hit that inhibited HIV-1 in a dose-dependent manner (Fig. 1A). Surface plasmon resonance (SPR) analysis indicated that H27 does not bind CA monomers or TRN-1 individually but does bind CA hexamers, albeit with low affinity ($K_D = 339 \pm 88 \times \mu M$, Appendix Fig. S2). The intermediate effect observed for compounds **6** and **29** proved to be artefactual in subsequent experiments as a result of overt toxicity at high concentrations.

To confirm the antiviral effect of H27 on HIV-1 infection, three models of infection were used: VSV-G pseudotyped HIV-1-Luc in HeLa-R5 cells, X4-tropic (NL4-3) and R5-tropic (BaL) HIV-1 infection of T lymphocytes (JLTR-R5 LTR-GFP). Inhibition was compared with two negative controls (compounds **31** and **40**). Results indicated that H27 had a dose-dependent antiviral effect irrespective of the cell type or mode of viral entry (Fig. 1B–D). Although the degree of inhibition appeared greater with VSV-G pseudotyped viruses, this only reflected the intrinsic differences in

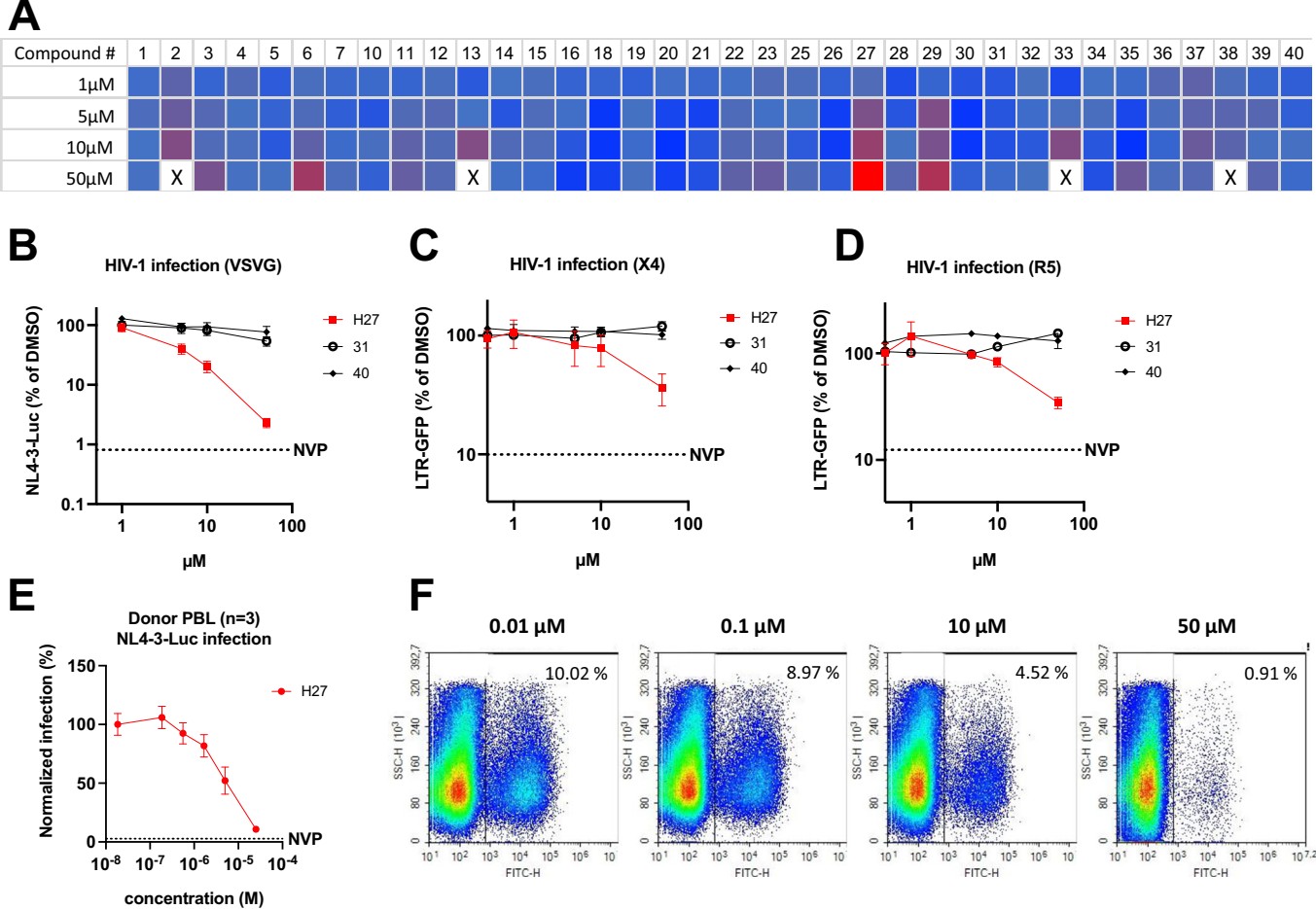

**Figure 1. H27 inhibits HIV-1 infection at low micromolar concentrations.**

(A) Heat map of the HIV-1 infectivity screen in HeLa-R5 cells of compounds selected by virtual screening. HeLa-R5-Luc cells were treated with compounds at the indicated concentrations for 2 h, then infected with VSV-G pseudotyped HIV-1-CMV-Nluc. Infectivity (NLuc) and viability (Luc) were assessed at 24 h post-infection (hpi). Results show mean NLuc/Luc values relative to DMSO control, and are representative of two independent screens. X = complete cell mortality. (B–D) Infectivity assays were performed in HeLa-R5 (B) or JLTR-5G (C, D) cells pre-treated with compounds at the indicated concentrations for 2 h, then infected with VSV-G pseudotyped LAI virus or in JLTR-5G with X4-tropic NL4-3 or R5-tropic BaL virus at 1 ng p24/1000 cells for 48 h. Results show the mean ± SD from three independent experiments. The Y-axes are on a logarithmic scale. (E, F) Primary PBL from healthy donors were pre-treated with H27 at the indicated concentrations for 2 h, then infected with NL4-3-Luc (E) or NL4-3-IRES-eGFP (F), and infectivity was assessed by plate luminometry at 48 hpi and flow cytometry at 72 hpi, respectively. In (E) curves show mean values from three different donors ± SEM. In (F) flow cytometry plots are representative of three independent experiments on three different donors. Source data are available online for this figure.

the dynamic range of the two assays (luminescence versus fluorescence) since all three viruses were equally sensitive to H27 when compared in the same system (HeLa-R5 cells, LTR-ßgal) (Appendix Fig. S3A). This antiviral effect was confirmed using replicative HIV-1 in primary peripheral blood lymphocytes (PBL), which are the natural target cells of HIV-1 infection in patients (Fig. 1E,F). $IC_{50}$ values obtained in cell lines and primary lymphocytes were compared with the previously reported CA inhibitors PF74 and LEN (Table EV1). The potency of H27 was 1.0–2.9 μM in HeLa-R5 and 2.9–5.6 μM in primary lymphocytes, which is in a similar range to PF74, whereas LEN displayed a very high potency, as previously reported (Bester et al, 2020; Link et al, 2020). In all assays, $CC_{50}$ (50% cytotoxic concentration) values for H27 were determined to be >100 μM, indicating a selectivity index ($CC_{50}/IC_{50}$) > 1000 (Appendix Fig. S3B,C; Table EV1).

## H27 inhibits HIV-1 nuclear import without disrupting reverse transcription

To determine the effect of H27 on the early steps of HIV-1 infection, we first performed quantitative PCR to measure reverse transcription, comparing the effects of H27 to control compounds **31** and **40**. The quantification of late-reverse transcripts at 7 hpi (Appendix Fig. S3D) and 24 hpi (Fig. 2A) indicated that H27 did not inhibit reverse transcription in any of the cell types that we tested, regardless of the mode of viral entry and the time post-infection.

To test the effect of H27 on HIV-1 nuclear import, we used three different approaches. For the first, we measured the nuclear entry of viral IN, which is imported with the viral PIC, using a protein complementation-based assay at 24 hpi (Fernandez et al, 2021). For the second, we used qPCR to measure 2-LTR DNA

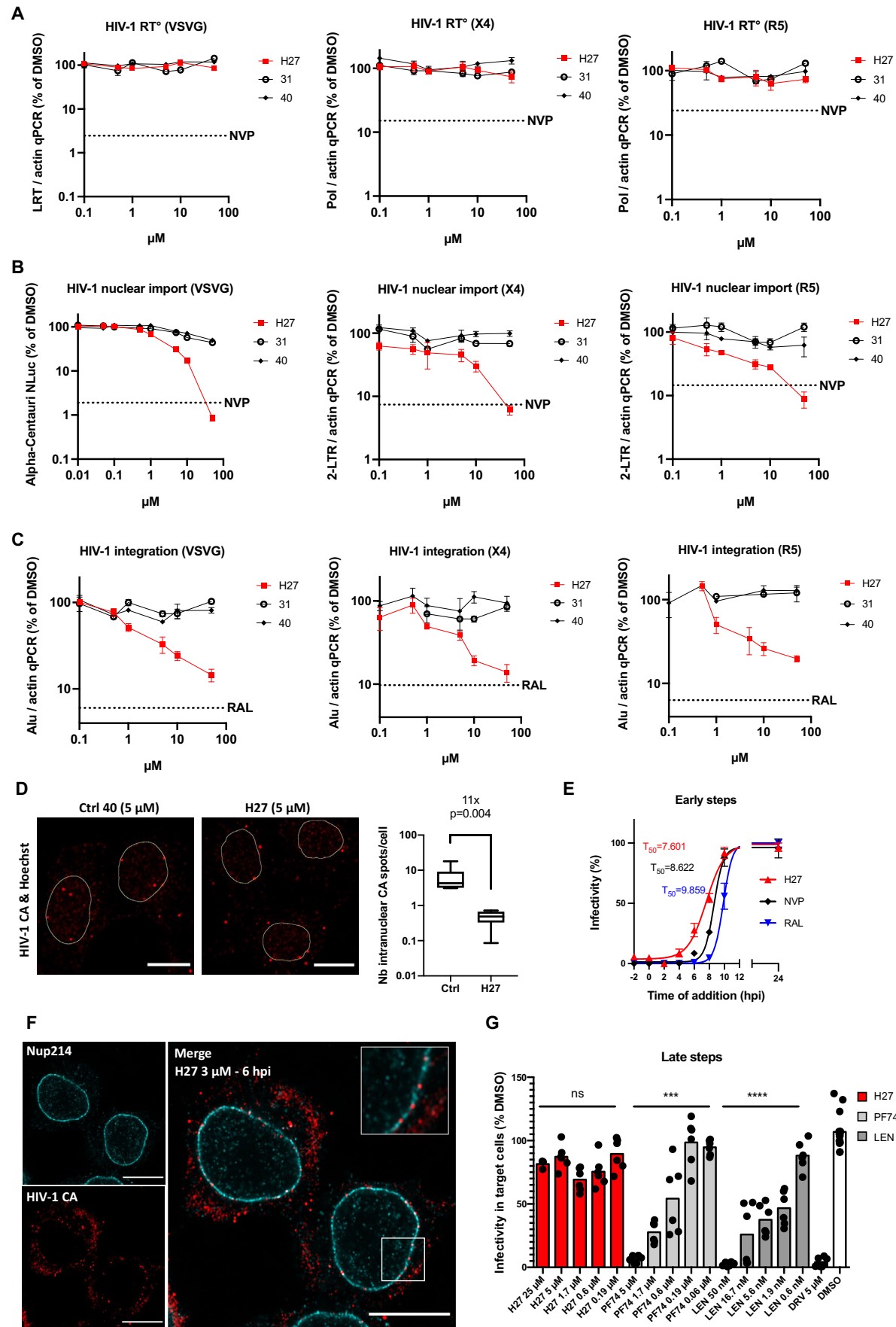

◀ **Figure 2.   H27 specifically blocks HIV-1 nuclear import.**

(A–C) HeLa-R5 cells were pre-treated with compounds at the indicated concentrations, then infected with VSV-G pseudotyped LAI, and JLTR-5G cells were infected with X4-tropic NL4-3 or R5-tropic BaL virus, at 1 ng p24/1000 cells for 24 h. Compounds 31 and 40 were included as negative controls since they do not affect HIV-1 infection. Results are the mean of four independent experiments ± SEM normalised for the corresponding DMSO control. The reverse transcriptase inhibitor NVP (5 μM) or the integration inhibitor RAL (10 μM) were included as a negative control in all experiments. (A) Reverse transcription (RT°) efficiency was assessed by qPCR amplification of late-reverse transcripts (LRT) or Pol gene and normalised for Actin values. (B) HIV-1 nuclear import in HeLa-R5 cells was assessed by Alpha Centauri protein complementation between a small NLuc fragment tagged C-ter of integrase (αHIV$^{NLuc}$) and a large NLuc fragment fused to a NLS (CenNLS$^{NLuc}$) (Fernandez et al, 2021). In JLTR-5G, HIV-1 nuclear import was assessed by qPCR amplification of 2-LTR circles and normalised for Actin values. (C) HIV-1 integration was assessed by qPCR and normalised for actin. (D) HIV-1 nuclear import was assessed by determining the localisation of CA at 6–8 hpi. Cells infected with LAI-VSV-G were fixed 6–8 hpi and labelled with anti-CA antibody and Hoechst. Confocal images acquired on an LSM800 microscope were quantified using Imaris on >300 cells from 4 independent experiments. Results are represented as a box and whisker plot with minimum to maximum bounds. Boxes extend from the 25th to 75th percentiles, while the line is plotted at the median. Statistical significance was assessed by unpaired t-test. Representative images with nuclei outlined in yellow are provided. Scale bar = 10 μm. (E) Time of addition assay in HeLa-R5 cells infected with VSV-G pseudotyped NL4-3-Luc and treated with compound 27 (H27, 5 μM), NVP (5 μM) or RAL (10 μM) at the indicated times pre/post-infection (hpi). The T$_{50}$ infectivity values indicate the times at which 50% of the viral complexes became resistant to the inhibitors. Results are the mean of two independent experiments performed in duplicate ± SEM. (F) HeLa-R5 cells were treated with H27 (3 μM) and then infected with LAI-VSV-G. Cells were fixed at 6 hpi, labelled with anti-CA and -Nup214 antibodies and stained with Hoechst. Images were acquired on an LSM880 Airyscan microscope with 63 × objective. Images are representative of three independent experiments (see Appendix Fig. S5 for full conditions). Scale bar = 10 μm. (G) To test the effect of H27 on HIV-1 assembly, HEK-293T cells were transfected with an NL4-3 molecular clone and treated with H27 at the indicated concentrations after 4 h. Supernatants were collected at 48 hpt and clarified by centrifugation. HeLa-R5 cells were then infected with diluted supernatants, and β-galactosidase luminescence was read at 48 hpi. Results show the mean from two independent experiments normalised for the DMSO control. Statistical significance was assessed using ordinary One-way ANOVA (Brown–Forsythe test).
****p ≤ 0.0001, ***p ≤ 0.001, ns non-significant. Source data are available online for this figure.

circles at 24 hpi, which are a standard marker of nuclear-localised HIV-1 DNA. Using both these assays, we found that H27 inhibited HIV-1 nuclear import (Fig. 2B), as measured by NanoLuc protein complementation at 24 hpi for VSV-G pseudotyped HIV-1 and qPCR at 24 hpi for R5- and X4-tropic viruses, to assess the nuclear entry of viral integrase and provirus, respectively. The decrease in nuclear import was confirmed by reduced levels of integrated HIV-1 provirus at 24 hpi, which were equivalent for all tested viruses, indicating an equal inhibition of nuclear import regardless of the envelope tropism (Fig. 2C).

For the third approach, we used confocal imaging to locate HIV-1 capsid at 6–8 hpi. HIV-1 capsid can enter the nucleus as a fully assembled capsid, therefore we verified that the AG3.0 anti-capsid antibody can recognise both monomeric and hexameric capsid. Using SPR, we confirmed that AG3.0 binds efficiently to both monomeric and hexameric capsid, with $K_D$ values of 4.12 nM and 53.9 nM, respectively (Appendix Fig. S4). Using this antibody to quantify cellular CA signal at 6–8 hpi revealed that intranuclear signal decreased 11-fold upon treatment with H27, from 3 to 18 spots per nucleus for the DMSO control to <1 spot per cell on average for H27-treated cells (Fig. 2D). The HIV-1 PIC that is transported into the nucleus contains minimally the viral genome, integrase and capsid (Hilditch and Towers, 2014). Therefore, since H27 inhibits the nuclear entry of all three components, these results suggested that H27 inhibits the nuclear transport of the HIV-1 PIC.

We next performed time-of-addition experiments to determine the window of pharmacological action of H27 compared with NVP and RAL, which inhibit reverse transcription and integration, respectively. Results indicated that 50% of the HIV-1 complexes became resistant to H27 at ~7.6 hpi, ~1 h prior to loss of sensitivity to NVP (~8.6 hpi), and two hours prior to RAL (~9.9 hpi, Fig. 2E), suggesting a block that occurred shortly before completion of reverse transcription and integration. This is concordant with recent work indicating that reverse transcription can occur after nuclear import (Burdick et al, 2020; Dharan et al, 2020; Li et al, 2021; Rensen et al, 2021; Selyutina et al, 2020).

## H27 treatment causes viral capsids to accumulate in the cytoplasm and at the nuclear envelope

Since recent work revealed that HIV-1 uncoating occurs in the nucleus (Burdick et al, 2020; Dharan et al, 2020; Li et al, 2021; Muller et al, 2021a; Selyutina et al, 2020; Zila et al, 2021), we next investigated whether H27 might, by inhibiting nuclear import, prevent HIV-1 uncoating. To address this, we quantified the capsid yield in infected cells by spinning cell lysates on a sucrose cushion, which allows intact capsids to be pelleted. Experiments were performed at 16 hpi, and in the presence of RAL to prevent the de novo synthesis of Gag. As expected for a nuclear import inhibitor, H27 led to a dose-dependent increase in pelletable CA (Fig. EV2A), and there was a concordant increase in capsid signal still present in infected cells at 16 hpi (Fig. EV2B).

To pinpoint where viral capsid accumulates with H27, we treated cells shortly before infection and imaged CA localisation at 2 and 6 hpi using high-resolution Airyscan and co-labelling NPCs with a Nup214 antibody. At 2 hpi, the CA signal appeared throughout the cytoplasm for all conditions, and in the nucleus for the DMSO control, as expected (white arrows, Appendix Fig. S5). At 6 hpi, the CA signal clustered at the cell periphery in the DMSO control, presumably corresponding to early assembly sites of de novo produced Gag. CA signal at the cell periphery was also observed for PF74 and LEN. In contrast, H27 caused the CA signal to remain in the cytoplasm and, strikingly, at the nuclear envelope, indicating a nuclear import block on capsids (Fig. 2F; Appendix Fig. S5). Control experiments performed on six different shuttling proteins (TRN-1, CPSF6, PML, IRF-3, NF-κB and influenza A virus nucleoprotein) indicate that H27 blocks the nuclear import of HIV-1 CA specifically (Appendix Fig. S6). These results are reminiscent of previous work showing the accumulation of HIV-1 particles at or near the nuclear membrane upon inducible nuclear pore blockade (Dharan et al, 2020), and confirm that the inhibition of HIV-1 nuclear import prevents viral uncoating.

## H27 does not inhibit the late assembly or maturation steps of HIV-1

Compounds that target the HIV-1 capsid, such as PF74 and LEN, are active at multiple stages of the HIV-1 life cycle, including early steps such as uncoating, nuclear import and integration, and late steps such as particle production and capsid assembly (Blair et al, 2010; McFadden et al, 2021). Therefore, we also assessed the effect of H27 on the late stages of the viral life cycle. Early steps were bypassed by transfecting cells with the NL4-3 molecular clone and treating virus-producer cells with H27, PF74, LEN or the protease inhibitor Darunavir (DRV). Infectious viruses in the supernatant were then titered on HeLa-R5 indicator cells (Fig. 2G). Results indicated that H27 did not block the production of infectious particles, whereas both PF74 and LEN had a profound and dose-dependent inhibitory effect. Therefore, unlike previously described CA inhibitors, H27 does not inhibit the late assembly or maturation steps of the HIV-1 life cycle.

## The inhibition of HIV-1 nuclear import by H27 analogues 43, 46, 49, 55 and 84 reveals a highly specific mode of action

To gain some insight into structure-activity relationships (SAR), forty-five commercially available analogues of H27 (compounds 41–85) were selected (Fig. EV3) and tested for their antiviral activity against HIV-1 in HeLa-R5 cells. Interestingly, only nine compounds (43, 46, 49, 55, 69, 80, 81, 84 and 85) retained an antiviral effect (Appendix Fig. S7). Moreover, compounds 69, 80, 81 and 84 displayed substantial cytotoxicity in HeLa cells (Appendix Fig. S7). We then assayed these analogues for their effect on nuclear import and confirmed a phenotype for five compounds (43, 46, 49, 55 and 84), which inhibited HIV-1 nuclear import in a dose-dependent manner, with $IC_{50}$ values in the low micromolar, or submicromolar range for compound 84 (Table EV2).

These results indicated that simplified analogues only featuring a N-(3-methoxy-4-(N-(4-methoxyphenyl)sulfamoyl)phenyl)benzamide component were as effective as H27 in these assays. The nature of the substituents on these benzamides turned out to be crucial since derivatives featuring fluorine atoms (in ortho, para positions, or both, compounds 55, 49 and 84, respectively) or chlorine (in para position, compound 69) were endowed with some effect. It is noteworthy that few examples of N-(3-(N-(4-methoxyphenyl)sulfamoyl)phenyl)benzamide also featuring such fluorine (in para, or para and ortho positions, compounds 80, 81, respectively) were active but showed significant cytotoxicity (Appendix Figure S7). The effect of the propyl-bearing analogue 85 or the sulfamide-bearing analogue 46 needs also to be considered, especially in light of the cytotoxicity noted for most of the active halogen-bearing analogues listed above.

These results also highlight the high degree of structural specificity required for activity. Indeed, while compound 43, which carries an ethoxy group in the para position, is antiviral in the same way as H27, compounds 72 and 83, respectively carrying a methoxy and propoxy group, lost this activity (Fig. EV3; Appendix Fig. S7). Since the loss or addition of a single methyl group led to a complete loss of antiviral activity, this suggests that the inhibition of nuclear import relies on precise and critical structural elements to ensure the antiviral activity of the analogues. This preliminary SAR pinpoints important contribution to some chemical groups and their positioning, but more molecules will be needed to draw conclusions on the role of single substituents or specific substitution patterns in terms of activity and cytotoxicity.

## H27 makes multiple contact points with HIV-1 CA hexamers

To identify CA residues that are important for H27 activity, we measured the effects of H27 on the CA-hexamer using highly sensitive methyl-TROSY NMR employing $^{13}C$ methyl labelled isoleucine and methionine (IM labelling). H27 was titrated into the IM-labelled CA-hexamer, and evaluation of these data showed that of the assigned side chain methyl resonances, the peaks corresponding to I2, M10, M39, I73, I91, I115, I150, I153 and M214 of HIV-1 CA showed very small chemical shift perturbation (Fig. EV4). Although this observation indicates that these residue sidechains experience a change in their local chemical environment upon ligand-binding, they are distributed over the entirety of CA and not located to a specific region or closely associated with the PR loop. This lack of delineation of a clear ligand binding site supports the notion that the inherent binding of H27 to individual CA-hexamers is weak. This is consistent with our SPR data and suggests that H27 may only bind productively to CA in the context of adjacent monomers when packed in the CA-lattice of assembled viral cores. It is entirely possible that H27 can bind at an intersubunit interface that is not present in isolated hexamers or CA monomers, such as the interhexamer trimer and dimer interfaces, or to areas of distinct curvature type across capsid shells, such as interfaces within pentamers or between pentamers and hexamers.

## Differences in the PR loop may account for differential sensitivity to H27 across divergent primate lentiviruses

To determine which CA residues within assembled capsid shells are critical for the antiviral activity of H27, we compared viral capsids from different primate species for susceptibility to H27. For this, we used previously described *gag-pol* constructs with the SIVmac251 background in which CA was replaced with that of either HIV-1 or different circulating SIVs (Mamede et al, 2013). Group M HIV-1, which is the pandemic form of HIV-1, resulted from cross-species transmission from chimpanzees, whereas HIV-2 originates from sooty mangabeys (Sharp and Hahn, 2011). These were compared to SIVs circulating in the wild: SIVmon isolated from a mona monkey, SIVgsn from a greater spot-nosed monkey, SIVagm from African green monkeys, SIVmac251, which was generated by inoculating macaques with SIV from sooty mangabeys, SIVmnd2 from mandrill and SIVcol from mantled guereza. All SIV chimeric constructs were tested in non-restrictive CHO cells in the presence of increasing concentrations of H27 or compound 31 as a control. Results indicated that capsids from SIVgsn and SIVmon were equally sensitive to H27 as HIV-1. In contrast, neither HIV-2, nor SIVmac, and none of the other CA from circulating SIV isolates tested here, including SIVcol, the most divergent SIV, were susceptible to H27 (Fig. 3A).

Primate lentiviruses show high sequence divergence, including in the *Gag* gene (Appendix Figure S8), however, evidence suggests that HIV-1 CA shares some structural similarities with CA from SIVgsn and SIVmon in the PR loop (Mamede et al, 2013), which

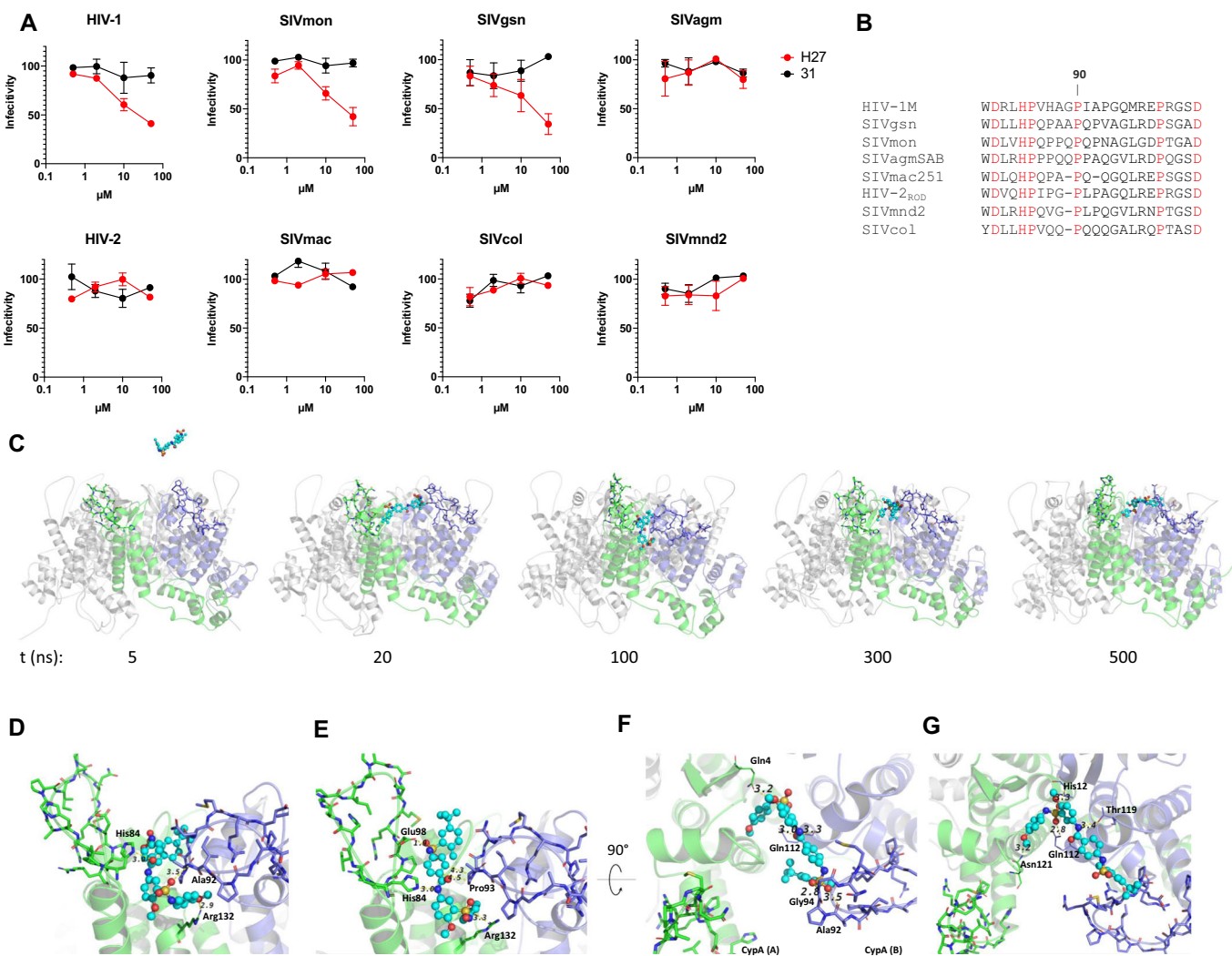

**Figure 3. Determinants for sensitivity to H27 across divergent lentiviral capsids.**

(A) CHO cells were infected with chimeric SIV constructs containing CA from diverse primate lentiviruses. Infection was assessed by GFP quantification, and results are represented as % of infectivity in 0.5% DMSO for each virus. Results show the mean of two independent experiments ± SEM. (B) Sequence alignments of the CypA-binding region of HIV and SIV CA used in Fig. 3A. Fully conserved amino acids are highlighted in red and gaps are indicated as "−". Full alignments of the CA are available in Appendix Fig. S8. (C) Snapshots from MD simulation with H27 (cyan ball & sticks) and CA hexamer (the two monomers involved in H27 binding are depicted in green and blue cartoon representation). H27 introduced into the proximity of CA selected a binding site located between two monomers near PR loops (residues depicted in sticks) after 20 ns and remained associated for the duration of the simulation (trapped in between two monomers). (D–G) Close-up views of the H27 binding site after (D) 100, (E) 150, (F) 300 and (G) 500 ns of simulation. Only hydrogen bonds formed between H27 and residues from the PR loop are shown for clarity. Additional Van der Waals contacts are observed with L83, H84, A92, P93, G94, P123, I124, P125). Source data are available online for this figure.

contains a PY-NLS that is recognised by TRN-1 (Fernandez et al, 2019). Sequence alignments of CA sequences revealed that HIV and SIV strains that were no longer sensitive to H27 had a slightly shorter loop (Fig. 3B). A similar phenotype was observed in terms of CA binding by CypA (Lahaye et al, 2013) and TRN-1 (Fernandez et al, 2019), for which the shorter loop in HIV-2 CA implies that key residues are not optimally positioned for binding to cellular co-factors.

We next conducted MD simulations in explicit water over 500 ns using the CA hexamer (PDB structure 3H4E) in the presence of the H27 compound located at about 30 Å above the hexamer. The aim of this study was to validate the predicted binding site obtained by docking but this time in the absence of TRN-1, reflecting better the experimental antiviral assays. In this way, the

molecule was free to move around the CA hexamer and to select preferential binding sites. After the equilibration step at constant temperature and pressure (310 K and 1 atm, 250 ps), the simulation was continued for another 500 ns (production phase used for analysis). The simulation indicated that H27 binds between two adjacent CA monomers within a hexamer at their NTDs (Fig. 3C). The binding trajectory was observed as three sequential stages, first H27 interacting with PR loops and making hydrogen bonds between residues H84, A92, P93, E98 and compound H27, then moving further inside close to α-helix-7 (R132) and finally residing stably bound to NTD near the α-helix-6 (Q112, T119 and N121, Movie EV1). It is of interest that residue E98 was shown to participate in the model of the complex formed by TRN-1 and the

CA hexamer (interaction with R813 making a salt bridge). This result suggests that H27 may destabilise the complex by weakening this interface of interaction. In addition to these hydrogen bonds, H27 binding made numerous van der Waals contacts throughout the simulation (Fig. 3D–G; Appendix Fig. S9), including I2, M10 and I115, which were also identified by NMR. Overall, residues that contacted H27 were located mainly between positions 79 and 132 of CA, spanning α-helices 4 to 7, and include nine residues within the previously described PY-NLS (H84 - R100, Appendix Fig. S9).

## PF74- and LEN-resistant mutants are sensitive to H27

To further probe the implication of CA residues in the antiviral effect of H27, we tested CA mutants for sensitivity to H27. The two common binding sites for host proteins and antivirals on HIV-1 capsid are the FG binding site, which is a hydrophobic pocket at the NTD-CTD interface of adjacent CA monomers, that is targeted by PF74, LEN, CPSF6 and TNPO3, and the PR loop, also called CypA-binding loop, that is targeted by CypA and TRN-1 (Engelman, 2021; McFadden et al, 2021). First, we found that H27 retained antiviral efficiency against CA mutants that were previously shown to escape recognition by the CA inhibitors PF74 or LEN (N74D, Q67H, and 5Mut (Q67H, K70R, H87P, T107N, L111I)) (McFadden et al, 2021; Shi et al, 2015) (Fig. 4A), confirming that it does not bind at the NTD-CTD interface that is targeted by PF74, LEN, CPSF6 and TNPO3 (Engelman, 2021). Antiviral activity was also retained, albeit with reduced efficiency, against the G89V mutant, which is consistent with the NMR and MD analyses, which did not identify G89 as a contact point for H27. These results suggest that H27 makes many additional contacts with the HIV-1 CA and the PR loop that are likely to minimise the importance of any one residue.

## Differential sensitivity of HIV-1 capsid mutants to H27

Next, we tested the sensitivity of CA mutants with altered stability or kinetics of uncoating (Saito and Yamashita, 2021). Most CA mutants have decreased infection efficiency, and in the case of Q219A and Q63A/Q67A, the block to infection was previously mapped to nuclear import or access to nuclear pores (Dismuke and Aiken, 2006; Yamashita et al, 2007). We found that the R143A and Q63/Q67A mutants, which destabilise capsids and can either accelerate or delay uncoating, were both sensitive to H27 (Saito and Yamashita, 2021), albeit at a fivefold higher $IC_{50}$ (Fig. 4B). A similar profile was obtained with the Q219A mutant, which destabilises cores and accelerates uncoating (Forshey et al, 2002; Saito and Yamashita, 2021). Intriguingly, the P38A mutant, which is also described as hypostable with accelerated capsid disassembly (Forshey et al, 2002), was more infectious in the presence of low concentration H27 (1 and 5 µM). Therefore, these variations in the degree of sensitivity to H27 may result from the fact that CA mutants not only affect capsid conformation and quaternary structure but can also have profound effects on the overall shell stability and also the degree of elasticity.

In vitro serial passage experiments with H27 identified two nonpolymorphic mutations, E45L and G46A and no other changes in CA. The E45L/G46A mutant was associated with viral outgrowth (Fig. EV5A,B), and resistance to H27 (Fig. EV5B,C). Interestingly, the E45L/G46A mutant was reminiscent of the previously described E45A hyperstable mutant (Forshey et al, 2002), which was also

resistant to H27, and was partially resistant to PF74 as previously shown (Shi et al, 2011) (Fig. EV5D). These results suggest that changes in capsid shell conformation, rigidity or elasticity may affect sensitivity to H27.

## Unlike LEN, H27 does not affect CA assembly or disassembly kinetics

Since H27 is a CA-targeting compound, we asked whether its antiviral effect might stem from its ability to modulate capsid assembly or disassembly, as has been reported for other CA inhibitors (Christensen et al, 2020; Faysal et al, 2024; McFadden et al, 2021; Selyutina et al, 2022). We, therefore, tested the effect of two antiviral analogues, H27 and compound 84, two controls (40 and 70) and LEN on the assembly and disassembly reactions of HIV-1 CA. Polymerisation in high salt was measured by monitoring turbidity at 350 nm as previously described (Hung et al, 2013) either in the absence or presence of compounds (40 µM final concentration). Results indicated that only LEN accelerated the assembly of HIV-1 CA (Fig. 4C). H27 did induce a slight slowing (twofold) in assembly kinetics, however this was also the case for all other tested analogues and controls (Table EV3). Similarly, the kinetics of particle dissociation indicated that LEN prevents disassembly but that no other tested compounds had any notable effect (Fig. 4D; Table EV3). Taken together, these data indicate that H27 has an antiviral mechanism that contrasts with previously described CA inhibitors, since it does not affect CA assembly or disassembly kinetics.

## H27 reduces HIV-1 CA binding to TRN-1

Since H27 was identified by virtual screening targeting the putative CA hexamer–TRN-1 interface, we considered that H27 could act as a PPI inhibitor, although other mechanisms cannot be excluded. We, therefore, attempted to determine whether H27 could interfere with capsid binding to NTRs such as TRN-1. Co-immunoprecipitation assays in cells transfected with HA-tagged TRN-1 indicated that H27 reduced the amount of CA pulled down by TRN-1 by about twofold when compared to DMSO or compound 31 (Fig. 5A). In addition, capsid pull-down assays, in which cores are isolated from infected cells and tested for the presence of endogenous TRN-1, indicated that H27 reduces the amount of TRN-1 associated with capsids by 2- to 10-fold in a dose-dependent manner (Fig. 5B). Moreover, SPR experiments confirmed that addition of excess H27 reduces TRN-1 association with immobilised CA hexamers by 50% (Fig. 5C).

To compare this phenotype with other CA-targeting compounds and other CA-binding proteins, we used Proximity Ligation Assay (PLA) in HeLa-R5 cells pre-treated with H27, PF74 or LEN and infected with HIV-1 VSV-G. Cells were fixed after 2 h and HIV-1 CA proximity to TRN-1 or CPSF6 were assessed by PLA (Fig. 5D). The H27 treatment reduced CA - TRN-1 PL spots almost to control levels from uninfected samples, while neither PF74 nor LEN had any effect at the dose tested (Fig. 5D,E). We did note a non-significant increase in signal with PF74, which may simply reflect increased CA epitope exposure (Francis et al, 2020). CA-TRN-1 PLA spots were detected both in the cytoplasm and the nucleus in the DMSO control, as previously published (Fernandez et al, 2019), and interestingly also in PF74- and LEN-treated samples, suggesting that nuclear import inhibition is not the main mode of action of

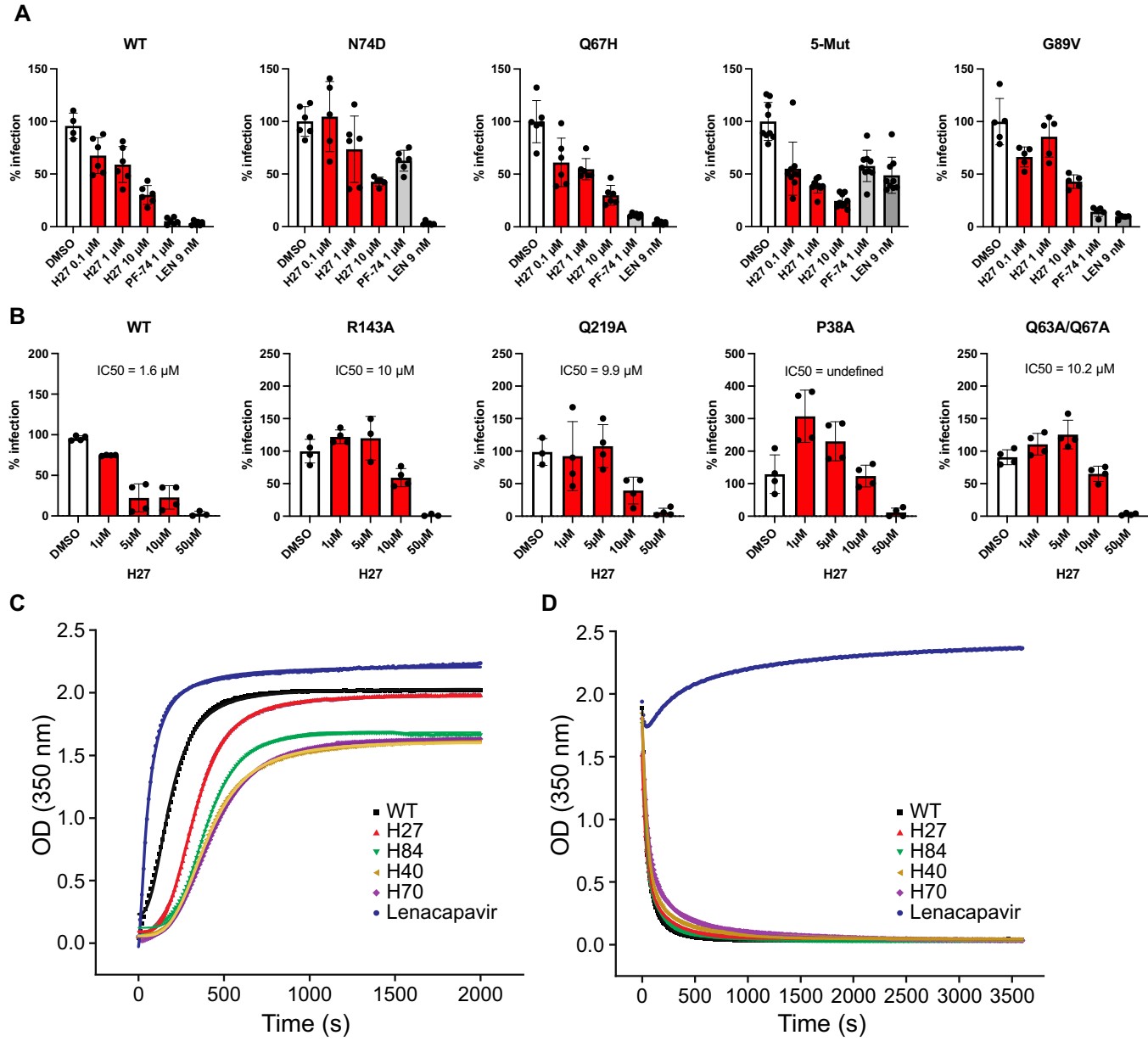

**Figure 4. Sensitivity of CA mutants to H27 and effect on CA assembly and disassembly.**

(A) HeLa-R5 cells were treated with H27 at 0.1, 1 or 10 μM, PF74 (1 μM) or LEN (9 nM) for 2 h, then infected with NL4-3 wild-type (WT), N74D, G89V or R9-GFP-VSV-G WT, Q67H and 5Mut (Q67H, K70R, H87P, T107N and L111I) at concentrations that yielded RLU values of ~2000 to normalise infections across mutants. β-galactosidase activity was measured at 48 hpi. Results are the mean of three independent experiments ± SD. Results obtained with NL4-3 and R9 viruses were similar and, therefore, merged in a single panel (left). (B) The effect of H27 on capsid stability R9 mutants was assessed in HeLa-R5 cells in the presence of increasing concentrations of H27 or compound **31** as control. Infectivity was assessed at 48 hpi by β-galactosidase assay. Results from two independent experiments were normalised for DMSO mean values in each dataset ± SD. IC$_{50}$ values were calculated by non-linear regression with variable slope (four parameters) using Prism 8. (C) Rates of HIV-1 CA assembly. Plots show the time dependence of OD$_{350}$ after the introduction of HIV-1 CA into high-salt assembly conditions in the presence of the indicated compounds. (D) Rates of HIV-1 CA disassembly. Plots show the time dependence of OD$_{350}$ after dilution of CA-assemblies from high-salt assembly conditions into a low-salt buffer in the presence of the indicated compounds. In each panel, a representative assay from three to four replicates is shown. The raw data were plotted as points, and the lines are the curve of best fit. Data from curve fitting and error analysis are given in Table EV3. Source data are available online for this figure.

these CA targeting molecules (Fig. 5E). In contrast, while LEN (9 nM) significantly reduced CA - CPSF6 PL spots, H27 (4.5 μM) had no effect, suggesting that H27 does not act at this interface (Fig. 5D). Of note, PF74 also failed to displace CPSF6 at 1 μM, which is concordant with others who reported that this may only be

achieved at high concentrations (15–25 μM) (Francis et al, 2020; Muller et al, 2021b), and therefore we cannot exclude that H27 might have a similar concentration-dependent effect on CPSF6.

Altogether, results suggest that H27 may act by preventing HIV-1 capsids from engaging with NTRs such as TRN-1, although

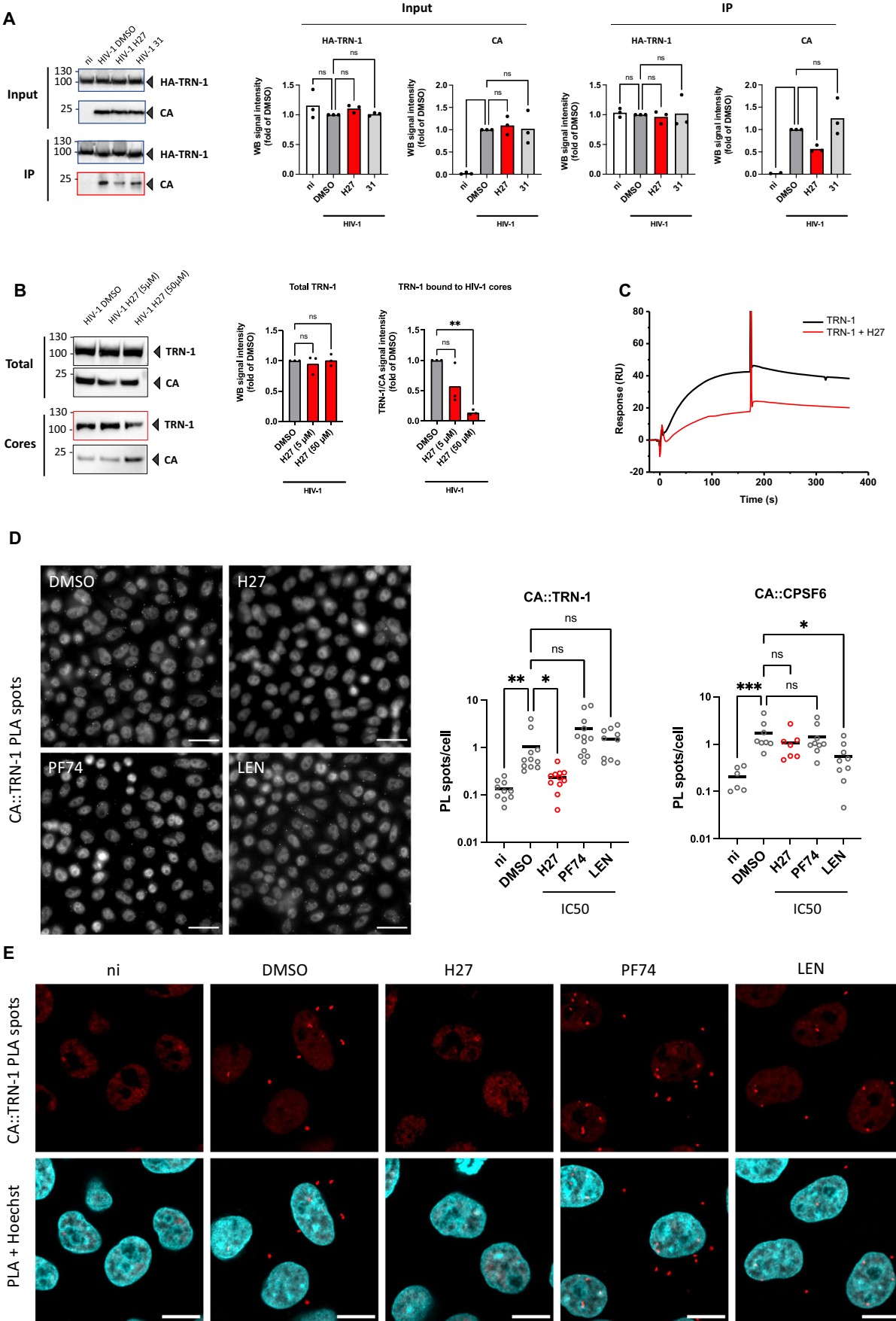

◄ **Figure 5. H27 inhibits the binding of HIV-1 cores to TRN-1.**

(A) Co-immunoprecipitation assay. HEK- 293T cells were transfected with pcDNA3.1 or pcDNA3.1-HA-TRN-1. At 24 h post-transfection (hpt), cells were treated with 50 μM H27 or compound **31** for 2 h, then infected with LAI-VSV-G for 6 h. Immunoprecipitation was performed using HA beads following hypotonic mechanical lysis. The bar graphs show the mean and scatter of the data from three independent experiments. Statistical significance was assessed by one-way ANOVA. (B) Capsid pull-down assay. HeLa-R5 cells were infected with LAI-VSV-G for 16 h in the presence of RAL to prevent de novo synthesis of Gag, and with 5 or 50 μM H27. Intact cores were isolated by ultracentrifugation on a 50% sucrose cushion, and TRN-1 co-sedimentation with capsids was assessed by Western blotting. The bar graphs show the mean and scatter of the data from three independent experiments. Statistical significance was assessed by one-way ANOVA. (C) Effect of H27 (500 μM) on the binding response of TRN-1 (1 μM) on CA-hexamer covalently immobilised on SPR sensor ship (4800 RU). (D, E) Proximity ligation assay. HeLa-R5 cells were infected with LAI-VSV-G at 1 ng p24/1000 cells in the presence of 4.5 μM H27, 1 μM PF74 or 9 nM LEN. Cells were fixed in 4% PFA at 2 hpi. The interaction between CA and either TRN-1 or CPSF6 in cultured cells was analysed by PLA (Duolink) using mouse anti-CA (AG3.0) and either TRN-1 or CPSF6 rabbit antibodies. Nuclei were stained with Hoechst. (D) Images were acquired using a Leica Thunder Imaging microscope with a 40 × objective. Representative images are shown. Scale bar = 30 μm. Dots indicate the number of PL spots per cell in a given field. Bars indicate mean values from a total of $n = 1900$–2300 cells per condition from three independent experiments. \*\*\*$p \le 0.001$, \*\*$p \le 0.01$, \*$p \le 0.05$, ns non-significant (Kruskal–Wallis test with multiple comparison). PL spots were counted using ImageJ software and analysed using PRISM GraphPad. (E) Samples from (D) were imaged using a Zeiss Airyscan microscope with a 63 × objective. Images are representative of three independent experiments. Scale bar = 10 μm. Source data are available online for this figure.

our data do not demonstrate that this is the mechanism responsible for the antiviral effect.

## Primary isolates from treatment-naïve patients and multi-resistant HIV-1 are inhibited by H27

To assess the translational potential of H27, we tested its antiviral efficacy on a panel of clinical samples from treatment-naïve patients. Plasma from 15 individuals with high viral load and covering eight different HIV-1 subtypes were obtained with patient consent (Table EV4). Viruses were isolated on primary PBMCs by culture over a 7-week period, and titration on HeLa-R5 cells indicated successful isolation for 7 of the 15 patient samples (Appendix Fig. S10), which were therefore selected to test their sensitivity to H27. For each plasma sample, we chose the earliest collection date that yielded detectable infectivity (~1000 RLU/s) and included a YU-2 control with equivalent infectivity. Results indicated that all clinical isolates were sensitive to H27 (Fig. 6A), suggesting that CA variations between HIV-1 clades are unlikely to confer natural resistance to H27.

Next, we tested HIV-1 mutants associated with common resistance to ART: IN-R263K, which confers resistance to dolutegravir (DTG), -G140S/Q148H and -N155H/K211R/E212T, which confer resistance to raltegravir (RAL); PR-I50V and -V32I/L33F and -I47V, which confer resistance to darunavir (DRV); RT-V106M, which confers resistance to the NNRTIs efavirenz (EFV) and nevirapine (NVP); and RT-M184V, which confers resistance to the NRTI lamivudine (3TC) (Malet et al, 2018) (https://hivdb-stanford-edu.proxy.insermbiblio.inist.fr/dr-summary/resistance-notes/NRTI/). Results indicated that all these mutants were sensitive to H27 (Fig. 6B), suggesting that H27 could retain some efficacy in treatment-experienced patients.

## In vivo pharmacokinetic analysis and assessment of in vivo toxicity of H27

To determine the pharmacokinetics of H27 in vivo, the compound was intravenously (i.v., Appendix Fig. S11A) or intraperitoneally (i.p., Appendix Fig. S11B) injected into male C57BL/6 mice at a 1 and 3 mg/kg dose, respectively. Sampling in the tail was performed from 5 min to 12 h, and from 15 min to 8 h post-i.v. and i.p. administration, respectively. The maximal plasma concentrations after a single injection of H27 was 1804–3197 ng/mL (2.8–5 μM) at 5 min after i.v. administration, and 130–894 ng/mL (0.2–1.4 μM) at

15 min after i.p. administration with the 3 mg/kg dose. The in vivo half-life of H27, which was estimated to be 1.6 h with i.v. route, was increased to 5.3 h with the i.p. route. We also noted that the plasma concentration of H27 remained unusually high (~70 ng/mL) at 8 h post-administration, indicating reasonable metabolic stability. Furthermore, the area under the curve (AUC 0–>24 h) was 1518 ng\*h/mL followed by i.p. route compared with 1021 ng\*h/mL by i.v. indicating a good exposure.

To quantify the possible in vivo toxicity of H27, 12 humanised mice were treated intraperitoneally with H27 ($n = 6$) or vehicle control ($n = 6$) every 2–4 days over 33 days at 3 mg/kg dose. Body weight (Appendix Fig. S11C) and clinical scores (Appendix Fig. S11D) were recorded on a weekly basis. Results indicated that body weight stayed stable over time in vehicle and H27-treated mice, except for one mouse treated with vehicle losing 19% of its initial body weight at D69 (Appendix Fig. S11C). Moreover, low average clinical scores (<2.5) were measured over time in both groups, except for one mouse in the vehicle group that presented an elevated score (=7) at D69. These results indicated no in vivo toxicity for H27 following intraperitoneal administration.

## Discussion

Nuclear localised HIV-1 genomes, either in the form of integrated or unintegrated DNA, account for the persistence of HIV-1 in patients despite successful antiretroviral therapy, and fuels viral rebound in case of treatment interruption. The current antiretroviral drugs that target reverse transcription or integration do not prevent a nuclear entry of the virus. Here, we identify a new class of HIV-1 inhibitors that specifically target HIV-1 nuclear entry and display antiviral activity. H27 was potent against all tested HIV-1, irrespective of their tropism, including HIV-1 isolates from the plasma of treatment-naïve patients and mutant viruses commonly associated with resistance to the four main classes of ART. Potency was demonstrated in primary human lymphocytes, with minimal toxicity and acceptable metabolic stability in vivo.

There is a general consensus that the HIV-1 capsid holds the key to viral access to the nucleus. It contains the determinants for transport across the nuclear pore in non-dividing cells (Yamashita and Emerman, 2004) and can persist as an intact core at the NPC, during passage through the NPC, and until uncoating in the nucleus (Arhel et al, 2007; Burdick et al, 2020; Dharan et al, 2020;

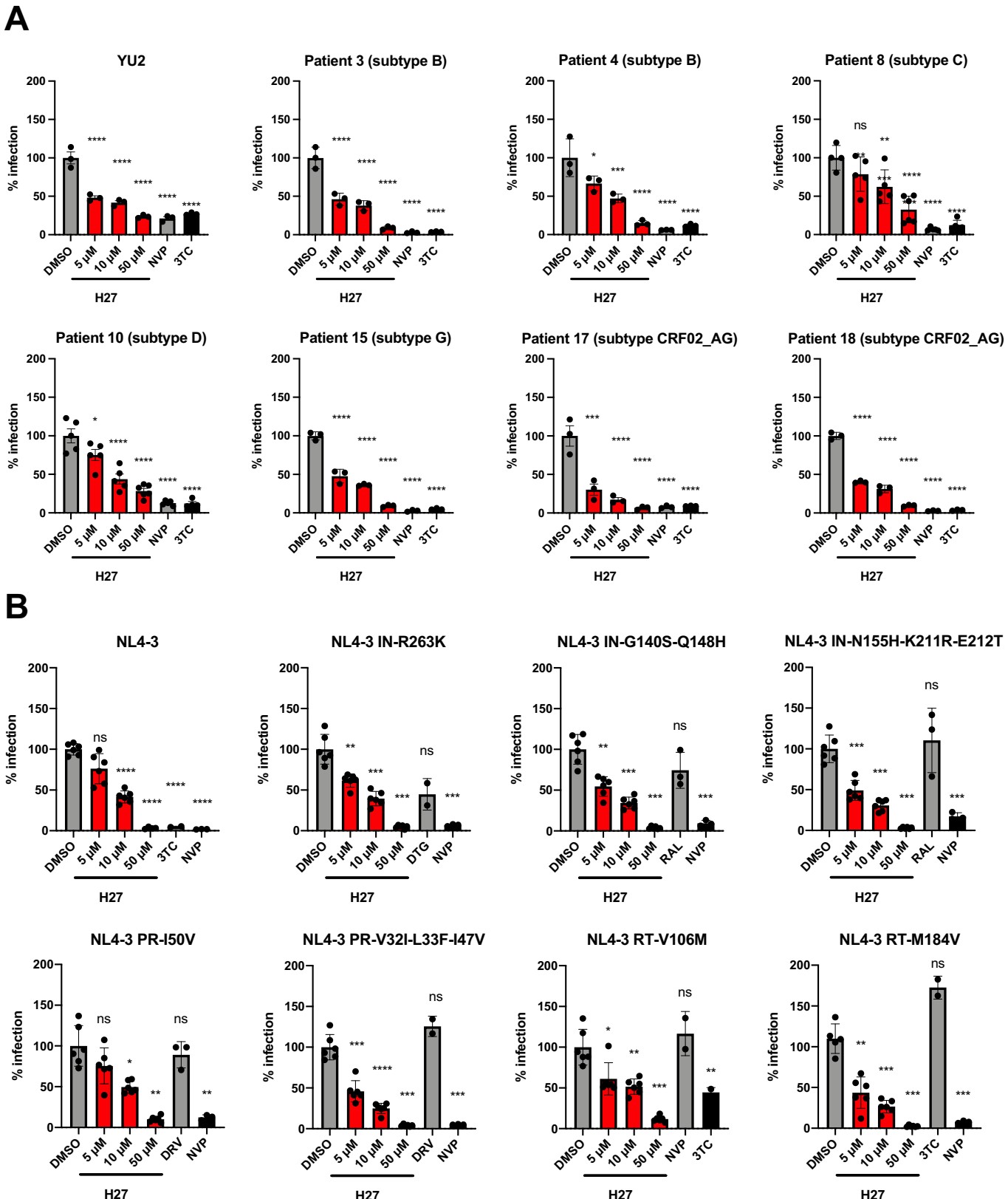

**Figure 6. Potency of H27 on HIV-1 isolates from treatment-naïve patients and resistant mutants to NRTIs, NNRTIs, PIs and INSTIs.**

(A) Sensitivity of clinical HIV-1 isolates to H27. The infectivity of different HIV-1 subtypes was tested on HeLa-R5 indicator cells in the presence of increasing concentrations of H27, 1 µM NVP or 5 µM 3TC and compared to the R5-tropic YU-2 molecular clone. The following collection dates from PBMC cultures (Appendix Fig. S10) were tested: d11 for patient 3, d32 for patient 4, d39 for patients 10, 15 and 18, and d42 for patients 8 and 17. Results are the mean of two independent experiments. (B) Effect of H27 on ART-resistant mutants. HeLa-R5 cells were pre-treated with the indicated concentrations of H27, then infected with NL4-3 replicative virus bearing mutations associated with clinical resistance to ART. Infection was assessed at 48 hpi by β-galactosidase activity using a plate luminometer. For each virus tested are included antiretroviral drugs to which it is insensitive and sensitive. Individual normalised values from two independent experiments are shown with mean ± SD. Statistical analysis was performed using Brown–Forsythe and Welch ANOVA tests with multiple comparisons to DMSO control. *$p < 0.05$, **$p < 0.005$, ***$p < 0.0005$, ****$p < 0.0001$, ns non-significant. Source data are available online for this figure.

Li et al, 2021; Muller et al, 2021a; Selyutina et al, 2020; Zila et al, 2021). Although the HIV-1 capsid is known to engage the nuclear import machinery in a number of ways, including by engaging with NTRs, with Nups and other cellular proteins that are karyophilic or alter its stability, the premise of our study was to use one such example to identify potential nuclear import inhibitors. Following on from our previous work (Fernandez et al, 2019), we virtually screened for compounds that are likely to bind the HIV-1 CA hexamer–TRN-1 interface. This allowed the identification of a new series of inhibitors that specifically block HIV-1 nuclear import.

H27's antiviral mechanism is distinct from previously described CA-targeting compounds since it did not alter CA assembly or disassembly kinetics and was effective against CA mutants that escape inhibition by PF74 or LEN. Although experiments on divergent lentiviral capsids indicated that HIV-2 and some SIVs were resistant to H27, suggesting that the determinants for sensitivity to H27 might be on the PR loop of CA, structural analyses revealed that H27 makes multiple weak contact points across CA. These may act to disrupt the ability of HIV-1 capsids to interact with components of the nuclear import machinery. Although we found that H27 reduces the binding of HIV-1 capsid to TRN-1, we cannot exclude that it also blocks other key interactions with other NTRs, Nups, or cellular co-factors important for nuclear import. Alternatively, or additionally, H27 binding to capsids may alter their flexibility, and hence their ability to pass through NPCs.

H27 did not bind to CA monomers and bound to CA hexamers with low affinity, which contrasts with previously described capsid inhibitors, PF74, GS-CA1 and GS-6207 (LEN), which bind to CA hexamers with a high affinity. Still, we cannot rule out that H27 can bind to more complex structures, such as capsid tubes or incoming cores in the context of infection. H27 may only bind productively to CA in the context of adjacent monomers when packed in the CA-lattice of assembled viral cores, or to areas of distinct curvature type across capsid shells, such as interfaces within pentamers or between pentamers and hexamers.

Additionally, H27 did not affect capsid assembly or disassembly, whereas multiple effects have been reported for PF74 and LEN, as a result of their propensity to disrupt normal capsid assembly and disassembly throughout the viral life cycle (Blair et al, 2010; Carnes et al, 2018; Christensen et al, 2020; Faysal et al, 2024; McFadden et al, 2021; Saito and Yamashita, 2021; Yant et al, 2019), often with opposite phenotypes at low (~1 µM) versus high (>5 µM) doses (Saito et al, 2016; Selyutina et al, 2022). As such, H27 does not behave as a CA inhibitor per se, but rather a CA-interfering compound that prevents proper engagement of the import machinery and *in fine* nuclear entry.

Although a delineated binding site on CA could not be identified, mutations that block the binding of compounds or proteins at the pocket situated at the NTD-CTD interface did not reduce the efficiency of H27. In contrast, the sensitivity of unrelated HIV/SIV capsids to H27 suggested that a common feature might be the length of the PR loop, which is another well-characterised binding interface of HIV-1 CA. Although we previously showed that in HIV-1, the glycine at position 89 is optimally positioned for binding to TRN-1 (Fernandez et al, 2019), G89V mutants were still sensitive to H27, confirming that H27 makes multiple contact points with the CA. Similarly, SIVagmSAB, which appeared to have the correct positioning of the glycine, was insensitive to H27. However, previous work suggested that the proline array, as well as the length of the PR loop, is critical for binding to cellular factors (Franke et al, 1994; Lin and Emerman, 2006), therefore the high number of prolines (PPPX$_2$PPX$_7$P) in SIVagmSAB may account for reduced sensitivity to H27. The complexity of contact points between H27 and HIV-1 CA may explain why the only escape mutations to H27 that were detected (E45L/ G46A) are likely to completely disrupt capsid integrity since infectious particles could not be generated. This is consistent with the fact that the E45A mutant, which was previously described as hyperstable (Forshey et al, 2002), was no longer sensitive to H27.

Together, our findings identify a new class of HIV-1 antiviral molecules that specifically inhibit HIV-1 nuclear import by targeting the viral capsid and blocking its access to the nuclear import machinery. Further work will be needed to precisely determine which interactions are preferentially blocked, and we do not exclude that TRN-1 - CA interactions might play a relatively minor role. This is suggested by the SPR data, which failed to detect a direct interaction between H27 and TRN-1. Protein interfaces are exciting targets for drug discovery (Chene, 2006), and H27 and its analogues constitute a new example of potential iPPIs (Ran and Gestwicki, 2018). The fact that H27 does not bind to cellular TRN-1 may also be favourable in averting a degree of toxicity in vivo. Indeed, H27 was well tolerated in mice, and CC$_{50}$ values were estimated to be above 100 µM in human primary lymphocytes. However, although good metabolic stability was observed in vivo, blood concentrations of H27 were lower than expected already at 5 min after i.v. administration (2.8–5 µM compared with a theoretical value of ~18 µM for the 1 mg/kg dose). Therefore, further developments towards improved chemotypes and formulations will be necessary to increase half-life, bioavailability and efficacy ahead of pre-clinical efficacy experiments. Encouragingly, the IC$_{50}$ and CC$_{50}$ values of H27 indicated a large pharmacological window, and the SAR already identified some important structural components that provided a 20-fold improvement of the IC$_{50}$, thus hinting at avenues of further optimisation of the molecular structure.

# Methods

### Reagents and tools table

| Reagent/Resource | Reference or source | Identifier or catalogue number |
|---|---|---|
| **Experimental Models** | | |
| HEK 293 T (*Homo sapiens*) | ATCC | CRL-11268 |
| HeLa (*Homo sapiens*) | ATCC | CCL-2 |
| HeLa-R5 (*Homo sapiens*) | Charneau et al, 1994 | P4 |
| HeLa-R5-Luc (*Homo sapiens*) | This study | P4-Luc |
| HeLa-E4 (*Homo sapiens*) | Fernandez et al, 2021 | E4 |
| CHO (Chinese Hamster Ovary) | ATCC | PTA-9816 |
| MDCK (Madin-Darby Canine Kidney) | Dr. Sandie Munier, Institut Pasteur | |
| JLTR-R5 (*Homo sapiens*) | NIH HIV Reagent Programme, Dr. Antoine Gross, CNRS | Derived from JLTR-5G |
| **Antibodies** | | |
| p24 Monoclonal (AG3.0) | NIH HIV Reagent Programme, Dr. Jonathan Allan | ARP-4121 |
| p24 Monoclonal (183-H12-5C) | NIH HIV Reagent Programme, Dr. Bruce Chesebro and Kathy Wehrly | ARP-3537 |
| SF2 p24 (antiserum, Rabbit) | NIH HIV Reagent Programme; produced by BioMolecular Technologies | ARP-4250 |
| mouse anti-TRN-1 | Abcam | ab10303 |
| rat anti-HA tag clone 3F10 | Roche | 11867423001 |
| mouse anti-β-Actin | Sigma | A1978 |
| rabbit anti-Nup214 | Abcam | ab84357 |
| rabbit anti-CPSF6 | Proteintech | 15489-1-AP |
| mouse anti-PML | Millipore | 05-718 |
| rabbit anti-IRF-3 | Proteintech | 11312-1-AP |
| rabbit anti-p65 NF-κB | Cell Signaling | 8242S |
| mouse anti-NP | BioRad | MCA400 |
| sheep anti-mouse HRP | Cytiva | NA931 |
| goat anti-rabbit HRP | GE Healthcare | NA934V |
| goat anti-rat HRP | Abcam | ab97057 |
| donkey anti-mouse Alexa 488 | Thermo Fisher | A-32766 |
| goat anti-mouse Alexa 555 | Thermo Fisher | A-21127 |
| goat anti-rabbit Alexa 555 | Thermo Fisher | A-21428 |
| mouse monoclonal anti-HA—Agarose (HA-7) | Sigma | A2095 |

| Reagent/Resource | Reference or source | Identifier or catalogue number |
|---|---|---|
| **Chemicals, Enzymes and other reagents** | | |
| Phytohemagglutinin | Thermo Fisher | 10082333 |
| interleukin-2 | Immunotools | 11340025 |
| Fugene-6 | Promega | E2691 |
| nevirapine (NVP | Sigma | SML0097 |
| raltegravir (RAL, | Sigma | CDS023737 |
| puromycin | Thermo Fisher | A1113803 |
| lamivudine (3TC) | | |
| darunavir (DRV) | | |
| dolutegravir (DTG) | | |
| Debio-025 | Debiopharm | |
| cyclosporin A | Selleckchem | S2286 |
| PF74 | Molport | T16500 |
| Lenacapavir (LEN | Molport | T11465 |
| **Software** | | |
| NAMD v2.14 and v3.0α8 | University of Illinois | |
| GOLD 2020.3 | Cambridge Crystallographic Data Centre | |
| Open Babel | | |
| DataWarrior | OpenMolecules | |
| PyMOL v2.5 | Schrödinger | |
| AlphaFold | Deepmind | |
| iControl | Tecan | |
| BIAevaluation software 4.1 | Cytiva | |
| LightCycler 480 | Roche | |
| Image Lab | BioRad | |
| NovoExpress | Agilent | |
| OriginPro | Microcal Software | |
| ImageJ | National Institutes of Health | |
| Imaris 64 8.3.1 | Oxford Instruments | |
| Prism 10 | Graphpad | |
| NMRPipe | NIST, US Dept of Commerce | |
| SPARKY | UCSF | |

## Cells

HEK-293T, HeLa, and CHO cells were cultured in Dulbecco's modified Eagle's medium-GlutaMAX supplemented with 10% foetal calf serum (FCS) and penicillin/streptomycin (all Gibco by Life Technologies). HeLa-R5 cells are CCR5-expressing LTR-β-gal indicator cells (Charneau et al, 1994). HeLa-R5-Luc (clone 15) are stably transduced with a lentiviral vector expressing Firefly Luciferase (LV-Luc) to monitor for cell number and

viability. HeLa-E4 cells are HeLa cells stably transduced with LV-CenNLS$^{NLuc}$, for infection by αHIV$^{NLuc}$ to perform αCentauri protein complementation assay (Fernandez et al, 2021). JLTR-R5 are a CD4 T cell line encoding GFP under the control of the HIV-1 promoter. All cell lines were tested on a regular basis for mycoplasma contamination. Primary T lymphocytes were isolated from peripheral blood from healthy donors (Etablissement Français du Sang) by Ficoll density gradient centrifugation. Cells were stimulated with 1 µg/mL Phytohemagglutinin for 1 day, then 10 ng/mL interleukin-2 for 3 days.

## Viruses and vectors

The following viruses were produced by transient transfection of HEK-293T cells by calcium chloride co-precipitation: X4-tropic NL4-3, NL4-3-Luc, NL4-3-IRES-eGFP and LAI either wild-type or Δenv and VSV-G pseudotyped, R5-tropic BaL and YU-2, R9 stability mutants (Chris Aiken, Vanderbilt University). For late-phase experiments, HEK-293T cells were transfected with FuGENE 6 Transfection Reagent (Promega) according to the manufacturer's instructions. The following lentiviral vectors were described previously: eGFP expressing Chimeric *gag-pol* constructs in a SIVmac251 backbone (Mamede et al, 2013), HIV-1 (Zennou et al, 2001) and HIV-2. NanoLuc was cloned by BamHI /XhoI digestion downstream of the CMV promoter to generate HIV-1-CMV-NLuc. Lentiviral vectors expressing an shRNA against TRN-1 and with internal CMV-eGFP (Fernandez et al, 2019) were titered on HeLa cells and used to transduce target cells at MOI 50. Knockdown was assessed by Western blotting and quantitative PCR (qPCR). Viruses and vectors were titered by Lenti-X-P24 rapid titre kit (Takara). The A/WSN/33 (H1N1) virus was kindly provided by Sandie Munier (Unité de Génétique Moléculaire des Virus à ARN, Institut Pasteur, Paris, France). It was produced by reverse genetics and amplified and titrated on Madin-Darby Canine Kidney cells (MDCK) cells.

## Antibodies

The following reagents were obtained through the NIH HIV Reagent Programme, Division of AIDS, NIAID, NIH: Anti-Human Immunodeficiency Virus 1 (HIV-1) p24 Monoclonal (AG3.0), ARP-4121, contributed by Dr. Jonathan Allan (dilution 1:250 for immunofluorescence, 1:1,000 for Western blotting), p24 Monoclonal (183-H12-5C), ARP-3537, contributed by Dr. Bruce Chesebro and Kathy Wehrly (1:1,000 for Western blotting), and Polyclonal Anti-Human Immunodeficiency Virus Type 1 SF2 p24 (antiserum, Rabbit), ARP-4250, contributed by DAIDS/NIAID; produced by BioMolecular Technologies (dilution 1:1,000 for Western blotting). Other primary antibodies were mouse anti-TRN-1 antibody (Abcam ab10303, dilution 1:100 for immunofluorescence, 1:1000 for Western blotting), rat anti-HA tag clone 3F10 (Roche 11867423001, dilution 1:1000 for Western blotting) and mouse anti-β-Actin antibody (Sigma A1978, dilution 1:3000 for Western blotting). Primary antibodies used for immunofluorescence were rabbit anti-Nup214 (Abcam ab84357, dilution 1:500), rabbit anti-CPSF6 (ProteinTech 15489-1-AP, dilution 1:50), mouse anti-PML (Millipore, 05-718, dilution 1:200), rabbit anti-IRF-3 (Proteintech, 11312-1-AP, dilution 1:100), rabbit anti-p65 NF-κB (Cell Signaling, 8242S, dilution 1:200) and mouse anti-NP (BioRad,

MCA400, dilution 1:1,000). Secondary antibodies for Western blotting used at dilution 1:3000 were sheep anti-mouse HRP (Cytiva NA931) and goat anti-rabbit (GE Healthcare NA934V) HRP. Goat anti-rat (Abcam ab97057) HRP was used at dilution 1:20,000. Secondary antibodies for immunofluorescence used at dilution 1:1000 were donkey anti-mouse Alexa 488, goat anti-mouse Alexa 555 and goat anti-rabbit Alexa 555 (Thermo Fisher). Mouse monoclonal anti-HA−Agarose antibody (HA-7, A2095 Sigma) was used for co-immunoprecipitation experiments.

## Virtual screening to identify potential hits interacting at the interface of TRN-1-CA hexamer

Prior virtual screening of chemical libraries by docking, the previously obtained TRN-1-CA hexamer complex (3D model obtained by rigid docking using two crystal structures, 4FQ3 for TRN-1 and 3H4E for CA (Fernandez et al, 2019) was subjected to an intensive conformational sampling (several heating and cooling cycles in explicit water) to evaluate its thermodynamic stability and to closely analyse the interface between both partners (especially near the PR loop). This step was achieved by following a simulated annealing protocol (from 310 K to 510 K by steps of 10 K for 20 ps and cooling down from 510 to 310 K using the same interval) before performing a production run (40 ns) and finally minimising the potential energy of the system using NAMD (Phillips et al, 2020) packages (v2.14 or v3.0α8) and the CHARMM36m force field (Huang and MacKerell, 2013; Huang et al, 2017). From this relaxed, energy-minimised complex, virtual screening was carried out using the GOLD programme (Genetic Optimisation for Ligand Docking, 2020.3 from the Cambridge Crystallographic Data Centre (Jones et al, 1997)) by targeting the main interface of the HIV-1 CA hexamer - TRN-1 complex around the PR loop of the CA protein and defined as a sphere of 10 Å of radius around T521 from TRN-1). Chemical libraries were prepared by filtering large databases (ChEMBL v25, $1.9 \times 10^6$ bioactive molecules, and MolPort $7.6 \times 10^6$ commercially available compounds mainly from Asinex, ChemDiv and Enamine) by using physicochemical criteria that best describe inhibitors of protein/protein interface (iPPIs). These were, namely, molecular weight between 400 and 1200 Da, ring content between 4 and 20, hydrogen bond acceptors between 4 and 20 and cLogP between −10 and 10. Initial filtering was performed using the Open Babel toolbox (O'Boyle et al, 2011) (for assigning the atomic Gasteiger-Marsili partial charges), and the final preparation of ligands was completed with the DataWarrior programme (Sander et al, 2015) for the removal of duplicates and generation of 3D coordinates. A total of 330,000 iPPI-like compounds were screened at the HIV-1 CA hexamer–TRN-1 interface using 20 genetic algorithms for the search of docking solutions. The ranking was achieved using the goldscore scoring function with the complete linkage clustering method from the RMSD matrix of generated solutions. For an overview of the global process, see Fig. EV1. From all screening runs, the first 40 best-ranked compounds (according to their docking score) were selected for purchase when commercially available (36 out of 40 were in stock) and further evaluated experimentally for their potential antiviral activities. This first series was supplemented by a second series consisting in highly similar structures of hit #27 and purchased from MolPort or ChemSpace resellers. All figures and analyses were carried out within the PyMol Molecular Graphics System (v2.5, Schrödinger, LLC).

## Molecular dynamics (MD) simulations of the CA hexamer with H27

Molecular dynamics simulations were performed to confirm the docking prediction and validate potential H27 binding sites on CA hexamer (PDB 3H4E) using NAMD as described above with GPU-based performances for non-bonded calculations and CHARMM36m as force field for protein chains supplemented by CGENFF-v4.6 (Vanommeslaeghe et al, 2010) for the topology and parameters of H27 molecule. CA hexamer–H27 complex was immersed in a water box (TIP3P model) with a 10 Å edge in each direction, neutralised by the addition of sodium and chloride ions at a physiological concentration (0.154 M) and replicated using periodic boundary conditions. The solvated system was energy minimised by performing 50,000 steps of conjugate gradients, gradually heated to 310 K and then equilibrated for a further 300 ps. The short-range Lennard-Jones potential was smoothly truncated from 10 to 12 Å and the PME (Particle Mesh Ewald) algorithm (Essmann et al, 1995) was used to calculate long-range electrostatics with a grid spacing of 1 Å. Newton's equation of motion was integrated using a timestep of 2 fs using the multiple-time-stepping integrator r-RESPA (Tuckerman et al, 1992), and constant pressure (1 atm) and temperature (310 K) were maintained in the isobaric-isothermal ensemble by means of Langevin dynamics and Nosé-Hoover Langevin piston (Feller et al, 1995). A trajectory of 500 ns was then produced (1 frame saved every ps) for further analysis. The same procedure was used for SIVmac CA starting from the best-ranked (and highest confidence) 3D model built by AlphaFold (Jumper et al, 2021).

## Infectivity assays

For infectivity assays in 96-well plates, cells were seeded the day prior to the experiment. On d0, cells were pre-treated with compounds and antiviral molecules for 2 h prior to infection, then infected with HIV-1 in 5% FCS-supplemented serum for the indicated times (hpi). Identical results were obtained by adding compounds simultaneously with the virus. Compounds were pre-diluted in DMSO to normalise all samples at 0.5% DMSO for all concentrations. Infection in 0.5% DMSO alone was used as a control and to normalise across infectivity assays. Infection was performed in 96-well plates at 0.5 ng p24/$10^6$ cells for HIV-1-CMV-NLuc for the initial HIV-1 infectivity screen of compounds identified by virtual screening, and with 1 ng p24/1,000 cells for all other infections with NL4-3, BaL or R9 viruses. All inhibitors were maintained during infection until read-out.

For all other infection experiments, and unless otherwise stated, cells were seeded the day prior to the experiment. On d0, cells were pre-treated with compounds and antiviral molecules for 2 h prior to infection, then inoculated with HIV-1 virus for 2 h in serum-free medium to promote cell attachment, and then incubated in 10% supplemented FCS for a further period of time referred to as hpi to allow infection to proceed. Unless otherwise stated, infections were all carried out at 1 ng p24/1000 cells (i.e. MOI 5, assuming that 1 ng of p24 corresponds to 5000 transducing units (Zufferey et al, 1998)).

Read-outs for infectivity assays in 96-well plates were as follows: HeLa-R5 were infected with VSV-G pseudotyped NL4-3-Luc and Luc signal was measured at 24 hpi by plate luminometry (Tecan Infinity M200) using a Bright-Glo luciferase assay substrate (Promega), or with

VSV-G pseudotyped LAI and β-galactosidase signal was measured at 48 hpi using a β-galactosidase reporter gene assay (Roche). Alternatively, JLTR-R5 were infected with replicative NL4-3 or BaL viruses and GFP was measured by flow cytometry (NovoCyte) at 48 hpi. The percentage of GFP-positive cells was 41.8 ± 7.6% for NL4-3 and 16.9 ± 1.6% for BaL in five independent experiments. Peripheral blood lymphocytes were infected with VSV-G pseudotyped NL4-3-Luc or NL4-3-IRES-eGFP and analysed by plate luminometry at 48 hpi, or by flow cytometry at 72 hpi, respectively.

$IC_{50}$ values were obtained using non-linear fit of log(inhibitor) vs. response with variable slope (four parameters) implemented in Prism 9. Toxicity was assessed by Bright-Glo Luciferase Assay System (Promega) for HeLa-R5-Luc cells, or by CellTiter-Glo luminescent cell viability assay (Promega) for all other cells.

## Surface plasmon resonance (SPR)

SPR experiments were performed on a Biacore T200 instrument (GE Healthcare) at 25 °C in in PBS running buffer containing 0.05% surfactant P20 (Cytiva) and 5% DMSO. For inhibition experiments, HIV-1 CA hexamer was diluted in 10 mM acetate (pH 5) and covalently immobilised (RU levels of 3000-5000RU) on a CM5S sensor chip by standard amine coupling according to the manufacturer's instructions. Binding experiments of recombinant TRN-1 (1 μM) to CA with or without H27 (500 μM) were performed at a flow rate of 30 μL.min⁻¹. The binding levels were compared after the end of the injection phase. The $K_D$ for H27 interaction with hexameric CA immobilised at 14,000 RU was performed by one-cycle kinetic titration at five increasing H27 concentrations (62 to 1000 μM, twofold dilution series) at a flow rate of 100 μL.min⁻¹. Sensorgrams were evaluated using a steady state fitting model (BIAevaluation software 4.1; Cytiva). All sensorgrams were corrected by subtracting the signal from the control reference surface (without any immobilised protein) and buffer blank injections before fitting evaluation. The experiments were repeated twice, with each time a new immobilisation of CA.

## Quantitative PCR (qPCR)

Cells were infected with 1 ng p24/1,000 cells. At 24 hpi, total cellular DNA was isolated using the QIAamp DNA micro kit (QIAGEN). The primer and probe sets used to detect each sequence are indicated in Table EV5. Reactions contained 1x Takyon Low ROX Probe MasterMix (Eurogentec), or LightCycler 480 Probes Master (Roche) for Alu-PCR, with 300 nM forward primer, 300 nM reverse primer, 100–200 nM probe primer and template DNA in a 10 μL volume. After initial annealing (50 °C for 2 min) and denaturation steps (95 °C for 10 min), 40 cycles of amplification were carried out (95 °C for 15 s, 60 °C for 1 min). All qPCR read-outs were normalised by amplification of the housekeeping gene β-Actin (Table EV5).

Assessment of integration was performed by a first amplification step using a U3-modified primer extended with the lambda phage-specific heel sequence in combination with two Alu primers, as previously described (Di Nunzio et al, 2012). The first-round PCR was then diluted 1:5 and amplified using primers and a probe for nested PCR (Table EV5). For the standard curve, DNA generated from 50,000 HIV-1 infected cells was endpoint diluted in Human Genomic DNA (Roche).

## Quantification of HIV-1 nuclear import by protein complementation assay (Alpha Centauri)

Hela-E4 stable Cen-nls cells (Fernandez et al, 2021) were treated with the indicated compounds for 2 h, then infected with LAI-α-VSV-G at 138 ng/20,000 cells in 96-well format. At 24 hpi, NanoLuc complementation indicative of viral nuclear import was measured using the NanoLuc substrate Z97 (Coutant et al, 2020; Coutant et al, 2019; Fernandez et al, 2021).

## Fate-of-capsid assay

HeLa-R5 cells were infected with VSV-G pseudotyped LAI in the presence of RAL to prevent de novo Gag expression. Cells were collected at 16 hpi and were lysed in hypotonic lysis buffer for 15 min at 4 °C. Cells were then disrupted with a Dounce homogeniser, and clear lysates (input) were collected after centrifugation at $1000 \times g$ for 4 min at 4 °C. Soluble CA and intact cores were separated by ultracentrifugation at $100,000 \times g$ for 2 h at 4 °C on a 50% sucrose cushion using an SW32 Ti swinging bucket rotor. Pellets were resuspended in 100 µL PBS for 90 min at 4 °C, before analysis by Western blotting.

## Western blotting

Cellular extracts were loaded onto 10% polyacrylamide gel and electrophoresis was performed in MOPS SDS running buffer (Invitrogen) in an Invitrogen chamber with a PageRuler Plus protein ladder (Thermo Scientific). The transfer was carried out in transfer buffer (25 mM Tris Base pH 8.3, 192 mM Glycine, 20% Ethanol) onto nitrocellulose membranes (Sigma). Incubation with primary antibody was performed in 5% milk powder overnight at 4 °C and secondary antibody for 1 h at room temperature. Horseradish peroxidase activity was revealed using Immobilon forte Western HRP substrate (Millipore). Images were acquired and analysed using Image Lab software (BioRad).

## Cell imaging and analysis

Cells were seeded onto 12 mm glass coverslips at least 1 day prior to fixation. Cells were fixed in 4% microscopy-grade paraformaldehyde (PFA, Alfa Aesar) for 10 min, permeabilised in 0.5% Triton X-100 for 15 min, then treated with 50 nM $NH_4Cl$ and 0.3% bovine serum albumin (BSA, Sigma) for 10 min each, to block the non-specific signal. Cells were incubated with primary and secondary antibodies at the recommended manufacturer dilution in 0.3% BSA for 1 h, or 30 min, respectively, in a wet chamber at room temperature. Coverslips were washed five times with PBS between each incubation. Finally, cells were stained with Hoechst 33342 (Invitrogen) for 5 min, and then mounted on Superfrost Plus glass slides (VWR) with ProLong Diamond (Molecular Probes). Proximity ligation assay was carried out with Duolink in situ PLA probe anti-mouse MINUS and anti-rabbit PLUS (Sigma) according to the manufacturer's protocol. Images were acquired on a Zeiss LSM880 confocal microscope with a 63 × objective, in Airyscan mode where indicated, or a Leica Thunder Imaging microscope with a 40 × objective for PLA experiments. Flow cytometry acquisition was performed using a NovoCyte flow cytometer and analysed using NovoExpress software (Agilent). Imaris 3D Surfaces and Imaris Spots (Oxford Instruments) were used to detect nuclei and CA clusters, respectively. Fiji was used to quantify PL spots.

## CA monomer expression and purification

The HIV-1 CA construct coding for residues 1–231 (reagent pWISP98-85) was obtained from the NIH AIDS Reagent Programme. The CA coding sequence was cloned into pET-11a between the NdeI and BamHI restriction sites. The protein was expressed in *E. coli* BL21 DE3 in 6 L (2 × 3 L) LB media and induced with 1 mM IPTG for 4 h at 30 °C. Cells were centrifuged at $8000 \times g$ resuspended in 70 mL of 50 mM Na-MOPS pH 6.9, 10 mM DTT and 1 mM Benzamidine and lysed by sonication. After removal of the cell debris by centrifugation for 45 min at $27,000 \times g$ proteins were precipitated for 1 h at 4 °C by adding an equal volume of ammonium sulfate to reach 50% saturation followed by centrifugation at $27,000 \times g$. Proteins were resuspended in 50 mM Na-MOPS pH 6.9 and 10 mM β-mercaptoethanol. Then one volume of 5 M NaCl was added, and the sample was incubated for 70 min on a rotating wheel at 4 °C and centrifuged at $20,000 \times g$ for 45 min. The pellet was resuspended in 20 mL, and another round of NaCl precipitation/spinning was performed. The pellet was finally dissolved in 20 mL of 50 mM Tris pH 8, 5 mM β-mercaptoethanol and dialysed against the same buffer overnight at 4 °C. The sample was loaded onto two 5 mL QFF columns connected in series and equilibrated with dialysis buffer. The protein was recovered in the flow-through after a wash step with 50 mM Tris pH 8, 50 mM NaCl and 5 mM β-mercaptoethanol. At this stage, about 43 mg of pure protein was recovered, concentrated to 5 mg.mL$^{-1}$, flash-frozen in liquid nitrogen and stored at −80 °C. The capsid was subjected to a final polishing step by injecting 5 mg of protein on a size-exclusion chromatography column (Superdex 200 10/300 GL increase column) and eluted in 50 mM Tris pH 8, 50 mM NaCl. The protein was concentrated to 4.6 mg.mL$^{-1}$, flash-frozen in liquid nitrogen and stored at −80 °C.

## CA mutant hexamer expression and purification

The CA A14C/E45C/W184A/M185A mutant (Pornillos et al, 2009) was synthesised with optimised codons for *E. coli* expression and cloned in pET-41 between the NdeI and XhoI restriction sites. The protein expression was performed in *E. coli* BL21 DE3 resistant to Phage T1 and co-transformed with the pRARE2 plasmid. Cultures were grown in LB media, and protein expression was induced with 1 mM IPTG. Bacteria from 3 L of culture were centrifuged and resuspended in 50 mM Na-MOPS pH 6.9, 200 mM β-mercaptoethanol and 1 mM Benzamidine and lysed by sonication. The lysate was centrifuged at $27,000 \times g$ for 35 min, and the supernatant was diluted with an equal volume of ammonium sulfate to reach 30% (w/v) saturation and left for 2 h at 4 °C under agitation. After centrifugation at $27,000 \times g$ for 45 min, the precipitated proteins were resuspended in ~15 mL of buffer 50 mM Tris pH 8, 200 mM β-mercaptoethanol, NaCl was added to a final concentration of 2.5 M NaCl final and left for 70 min at 4 °C. After a centrifugation step at $27,000 \times g$ for 45 min the same NaCl procedure was applied. Finally, the protein pellet was resuspended in 20 mL of 50 mM Tris pH 8 and 200 mM β-mercaptoethanol. The solubilized proteins were loaded on two 5 mL QFF connected in

series and equilibrated with 20 mL of 50 mM Tris-HCl pH 8 and 200 mM β-mercaptoethanol. A wash step with 50 mM Tris pH 8 and 200 mM β-mercaptoethanol was applied, and most capsid protein was eluted in the flow-through and wash steps. The protein was then concentrated to about 25 mg.mL$^{-1}$ and dialysed against 50 mM tris pH 8, 200 mM β-mercaptoethanol, 1 M NaCl O/N at 4 °C then 8 h against 50 mM tris pH 8, 0.2 mM β-mercaptoethanol, 1 M NaCl at 4 °C and finally O/N dialysis against 50 mM Tris pH 8 at 4 °C. The successful hexameric capsid reconstitution was assessed on a size-exclusion chromatography column (Superdex 10/300 GL increase column) in 50 mM Tris pH 8, almost all the protein eluted at in these conditions at 12.5 mL as a single monodisperse peak. For NMR experiments requiring IM methyl-isotope labelling, protein expression and purification of HIV-1 CA hexamer has been described (Nicastro et al, 2022). Briefly, cells were grown in minimal media containing $^{15}$NH$_4$Cl and [$^2$H,$^{12}$C]-glucose in 99.9% $^2$H2O. For the production of U-[$^2$H],Ile-δ1-[$^{13}$CH3] samples, 70 mg l$^{-1}$ of alpha-ketobutyric acid (methyl-$^{13}$CH$_3$) was added, and for [U-2H], ε-[$^{13}$CH$_3$]-Met labelled samples 200 mg l$^{-1}$ of [$^{13}$CH$_3$]- methionine was added to the culture at 1 h prior to IPTG induction.

## CANC and TRN-1 expression and purification

The purification of HIV-1 CANC and TRN-1 was described in a previous publication (Fernandez et al, 2019). TRN-1 was cloned into pGEX-6P1 in a frame and C-terminal to a glutathione-*S*-transferase- (GST) and hexahistidine-tag. A Tobacco Etch Virus protease recognition peptide was added after the Tags, enabling to remove them during the purification process. TRN-1 was expressed in *E. coli* BL21(DE3) resistant to phage T1 (New England Biolabs) transformed with the pRARE2 plasmid. An O.N. pre-culture was grown at 37 °C in a Luria-Bertani (LB) medium containing 200 µg/mL ampicillin and 30 µg/mL chloramphenicol. The starter culture was used to inoculate four flasks of 3 L LB with antibiotics at 37 °C media, when OD at 600 nm reached 0.8 the flasks were placed in melting ice for 30 min and protein expression was induced by the addition of 1 mM isopropyl β-D-1-thiogalactopyranoside (IPTG). After induction, the culture were grown O/N at 18 °C. Bacteria were centrifuged at 6000 × *g*, and the pellet was resuspended in 120 mL of buffer A: 50 mM Tris-HCl pH 8, 400 mM NaCl, 5 mM β-mercaptoethanol, 10% glycerol and 1 mM benzamidine. Cells were sonicated, and the lysate was clarified by centrifugation (27,000 × *g*, 4 °C, 45 min). The lysate was incubated O/N at 4 °C with 5 mL of GST-Sepharose beads (Cytiva) under agitation. The sample and beads were then loaded into a gravity column and washed with 5 column volumes (CV) of buffer B: 50 mM Tris-HCl pH 8, 1 M NaCl, 5 mM β-mercaptoethanol, 10% glycerol. The protein was then eluted with 5 CV of GST elution buffer: 50 mM Tris-HCl pH 8, 0.2 M NaCl, 10% glycerol and 20 mM reduced glutathione. At this stage, 250 mg of total protein was obtained. The protein was cleaved with 9 mg of hexahistidine-tagged-protease from Tobacco Etch Virus (TEV) during dialysis at 4 °C in 50 mM Tris-HCl pH 8, 0.1 M NaCl, 5 mM β-mercaptoethanol and 10% glycerol. The sample was then purified further by affinity chromatography (IMAC) (Ni-NTA Sepharose, GE Healthcare Life Sciences). This step allowed to separation of the cleaved Tags and the TEV protease that bound to the IMAC beads while the cleaved TRN-1 (about 90% of the sample) was collected in the flow-through. The protein was then concentrated into Vivaspin20 (50,000 MWCO, Sartorius) to 6.3 mg/mL, aliquoted in 1.8 mL tubes and flash-frozen in liquid nitrogen. At this stage, about 100 mg of very pure protein was obtained. Prior to its use for biochemical assays, TRN-1 was further purified on size-exclusion chromatography (Superdex 200 10/300 GL column, Cytiva) and eluted with a buffer made of 50 mM Tris-HCl pH 8, 0.2 M NaCl, 5 mM β-mercaptoethanol and 10% glycerol. Pure fractions corresponding to monomeric TRN-1 were collected and concentrated into Vivaspin20 (50,000 MWCO, Sartorius). After this three-step purification procedure TRN-1 was estimated to be as pure as 95% as judged on Coomassie stained SDS-PAGE.

## NMR spectroscopy

For NMR studies, H27 was dissolved in deuterated dimethyl sulfoxide (d$_6$-DMSO) at a stock concentration of 25 mM. The IM $^{13}$C methyl-labelled CA-hexamers were prepared by dialysis into NMR buffer (100 mM NaCl, 50 mM NaPO$_4$ pH 7.2, 100% D$_2$O). $^1$H-$^{13}$C HMQC methyl-NMR were recorded at 298 K on a Bruker Avance spectrometer operating at 700 MHz $^1$H frequency, and spectra were processed using NMRPipe and Sparky within the NMRbox virtual environment (Maciejewski et al, 2017). Resonance assignments of the Ile δ1- and Met ε-methyl groups in the CA-hexamer were obtained from the comparison of methyl-methyl nuclear Overhauser effects (NOEs) with the network of short methyl-methyl distances observed in the CA-hexamer crystal structure (PDB:3H47) as described (Nicastro et al, 2022). For IM chemical shift mapping, 50 µM of $^{13}$C IM-labelled HIV-1 CA-hexamer was titrated with H27 at increasing concentrations of 43 µM, 130 µM, 216 µM and 1 mM. The titration was monitored by acquiring a series of 2D $^1$H-$^{13}$C HMQC-TROSY spectra. Chemical shift perturbations were calculated according to the equation: $(\Delta\delta = \sqrt{(\Delta\delta H/\alpha)^2 + (\Delta\delta c/\beta)^2})$ (Rosenzweig et al, 2013), where $\Delta\delta_X$ is the chemical shift change of the nucleus X (H or C) between the two states and the parameters $\alpha = 0.29$ (Ile Hδ1), 0.40 (Met Hε); $\beta = 1.66$ (Ile Cδ1), 1.82 (Met Cε) are the standard deviations of the $^1$H and $^{13}$C chemical shifts for the specified atom taken from the Biological Magnetic Resonance Data Bank.

## Chimeric *gag-pol* SIVmac constructs

Chimeric *gag-pol* SIVmac expression plasmids were described previously (Mamede et al, 2013). CA sequences were SIVmon (accession No. AY340701) isolated from mona monkey (*Cercopithecus mona*); SIVgsn (AF468658) from greater spot-nosed monkey (*Cercopithecus nictitans*); SIVagmSAB1 (S46346) from African Green Monkey (*Chlorocebus sabaeus*); SIVcol (AF301156) from mantled guereza (*Colobus guereza*). SIVmnd2 CA was cloned from plasma from SIVmnd2-infected mandrills (*Mandrillus Sphinx*) (Onanga et al, 2006). The SIVmac251 sequence was obtained from a *gag-pol* SIVmac251 expression plasmid (pAd-SIV4) (Negre et al, 2000). Clustal (Omega) and Multalin (Corpet, 1988) were used to align sequences.

## Kinetics of CA assembly and disassembly

The effect of compounds (H27, H40, H70, H80 and Lenacapavir) on the assembly and disassembly reactions of HIV-1 CA-WT was measured by monitoring turbidity at 350 nm. For assembly,

purified CA-WT protein was dialysed into 50 mM $Na_2HPO_4$ at pH 7.5 and polymerisation triggered by the addition of 50 mM $Na_2HPO_4$, 5 M NaCl at pH 7.5. to give a final concentration in assembly reactions of 36 μM CA-WT protein and of 2.0 M NaCl, either in the absence or presence of compounds (40 μM final concentration). After the addition of NaCl to initiate assembly, the reaction was rapidly mixed and placed into a cuvette. Approximately a delay of 10 s elapsed between the time of addition and the first time point measured. The increase in optical density was monitored on a JASCO V-760 spectrometer at 350 nm every 10 s for 1 h at 25 °C.

For kinetic analysis of CA-WT disassembly, CA-WT polymers assembled as described above were prepared and incubated for 2 h before triggering the disassembly reaction by diluting tenfold into reaction buffer (50 mM $Na_2HPO_4$ at pH 7.5), in the presence of compounds (40 μM), with final concentrations of CA-WT and NaCl of 36 μM and 200 mM, respectively. The kinetics of particle dissociation triggered by dilution was determined at 25 °C by following the decrease in the optical density at 350 nm, as a measure of the remaining light-scattering particles. Data points were collected every 10 s for 1 h, in the presence of stirring at 250 rpm.

To estimate the rate of assembly or disassembly, the time-dependent increase or decrease in turbidity, in optical density at 350 nm, at early times was fitted to an empirical Hill function or exponential functions, respectively, by using OriginPro software (Microcal Software, Inc.) (see Table EV3).

Stock solutions of H27, H40, H70 and H80 were prepared by dissolving them in 100% dimethyl sulfoxide (DMSO) to a final concentration of 50 mM and Lenacapavir to a final concentration of 20 mM.

### Resistance mutations under selective pressure by H27

JLTR-R5 cells were inoculated with replicative NL4-3 virus at MOI 0.1 in the presence of H27, or DMSO and NVP as negative and positive controls. Viral replication was monitored every 3–4 days by GFP expression and compound concentrations were increased in case of GFP expression that was above the NVP control. H27 was increased by increments of 2–3-fold, starting with 0.2 μM on day 1, until 10 μM from day 34 onwards. Supernatants were then tested for infectivity and sensitivity to 10 μM H27 in HeLa-R5 cells, and supernatants at D49 from both DMSO and H27-treated cultures were selected for sequencing. RT-PCR products were obtained by RNA extraction, reverse transcription using PrimeScript RT enzyme (Takara), and PCR was performed to amplify Gag using high-fidelity Platinum Taq (Invitrogen). Direct bidirectional-Sanger sequencing was then performed on Topo clones. The E45L/G46A substitutions were introduced into the pNL4-3 sequence by site-directed mutagenesis using the site-directed mutagenesis kit QuickChange II XL (Agilent), according to the manufacturer's instructions. The E45L/G46A virions were virtually non-infectious, and were therefore produced by co-transfecting wild-type pNL4-3 with a pNL4-3/pNL4-3 E45L/G46A ratio of 1:9 to yield infectious particles.

### Co-immunoprecipitation assay

HEK-293T cells were seeded in 100 mm culture dishes (5 × $10^6$ cells) on day 0, transfected with 15 μg pcDNA3.1+ or pHA–TRN by calcium phosphate precipitation on day 1, and treated with compounds then infected with LAI-VSV-G virus (40 μg p24) on day 2. At 6 hpi, cells were trypsinized, and cell pellets were incubated in 500 μL hypotonic lysis buffer (10 mM Tris-HCl, pH 8, 10 mM KCl, 1 mM EDTA and protease inhibitor cocktail, Roche)) overnight at 4 °C. Cells were lysed mechanically by dounce homogeniser then centrifuged for 4 min at 1000 × g at 4 °C to pellet cell debris. A 50 μL aliquot from the supernatant was kept as an input sample, and the remaining supernatant was used for HA immunoprecipitation by adding 100 μL of anti-HA-Agarose antibody overnight at 4 °C on a wheel. The beads were washed twice in the lysis buffer before elution in 50 μL 2x Laemmli buffer, then vortexed and heated for 10 min at 95 °C. Finally, a 5 min centrifugation at 8000 × g was performed to pellet the beads and the supernatant was transferred to a new tube and stored at −20 °C before western blot analysis.

### Patients and plasma collection

Plasma from HIV-1 infected and antiretroviral treatment-naïve patients was obtained from an existing database. The patients had been informed that their demographic and clinical data would be recorded during their follow-up and that their peripheral blood samples would be stored and could be used for retrospective studies. Informed consent was obtained from all human subjects and confirmed that the experiments conformed to the principles set out in the WMA Declaration of Helsinki and the Department of Health and Human Services Belmont Report. Fifteen plasma samples were collected from patients who were diagnosed HIV-1 positive in the Pitié-Salpêtrière hospital.

### Amplification of primary isolates from HIV-1 patient sera

HIV was isolated from patient plasma based on a previously published protocol (Ehrnst et al, 1988). On day 0 (d0), all 15 serum samples were ultracentrifuged at 55,000 × g for 1 h, and pellets were resuspended in 700 μL RPMI containing 4 × $10^6$ activated PBMCs in the presence of DEAE-dextran. After incubation at 37 °C for 30 min, cells were transferred to a 12-well plate in a total volume of 2 mL RPMI medium supplemented with IL2. On d2, cells were transferred to a T25 for the remainder of the experiment, in total culture volumes that were gradually increased (4 mL on d2, 9 mL on d4, 15 mL on d13). One-third of the volume was harvested at 3–4 day intervals as of d7 over a 7-week period. Fresh autologous activated PBMCs were added on d11 and d35. Supernatants from infected PBMCs were titered by infection of reporter LTR-β-gal HeLa-R5 cells using a Tecan plate-reader to determine β-galactosidase luminescence after 48 h infection.

### ART-resistant HIV molecular clones

Mutations were introduced in the main target genes of antiretrovirals, R263K, G140S-Q148H and N155H-K211R-E212T in the integrase gene, V106M and M184V in the reverse transcriptase as well as I50V and V32I-L33F-I47V in the protease gene. These various substitutions were introduced into the pNL4-3 sequence by site-directed mutagenesis using the site-directed mutagenesis kit QuickChange II XL (Agilent), according to the manufacturer's instructions. After mutagenesis, all the plasmids were checked by sequencing at the level of the mutated gene.

## Animals

Male C57BL/6 mice were housed at Pharmacelsus animal facilities, with the approval of the Official Veterinary Service of the State Office of Saarland for Consumer Protection (reference 2.4.2.2/14-2020). Animals were fed *ad libitum* before and during the study and were inoculated by intravenous route with 1 mg/kg and 3 mg/kg of H27 formulation or solely the vehicle as control. Mice were euthanised by carbon dioxide inhalation as recommended by the European guidelines for the welfare and euthanasia of small rodents. Female NOD/Shi-scid/IL-2Rγnull immunodeficient mouse strain (NCG) were generated using hematopoietic stem cells (CD34+) isolated from human cord blood and were housed by Transcure Bioservices in accordance with TCS Standard Operating Procedures. Mice were 30 weeks old at the start of the experiment. For each animal study, mice were randomly allocated in the treated and control groups. No blinding was established for the experiments. No animals were excluded from the analysis.

## Formulation of H27 for injection in mice

The selected doses (1 and 3 mg/kg) were adapted to obtain an exposure equivalent to 6 and 20 times, respectively, the $IC_{50}$ value obtained in primary human T cells. The solution for intravenous administration was prepared by diluting the stock solution of the H27 compound in DMSO at 50 mM in sterile water and Kolliphor HS15 (Sigma-Aldrich/Merck) so the final concentration was 1 mM with a ratio of 2/30/68 (DMSO/Kolliphor HS15/H20). The same components were used to infect this formulation (vehicle) in the control mice group. HPLC analysis confirmed that H27 solubilised successfully in this formulation.

## Detection of H27 in blood samples by liquid chromatography and mass spectrometry (LC-MS)

Blood samples (20 μL) were collected from the tail with capillaries coated with K3-EDTA. Samples analysis was performed by LC-MS to quantify H27 and estimate the pharmacokinetic parameters. Within-run accuracy was determined as follows: samples with known concentrations (quality control, QC) of H27 were analysed by LC-MS and measured value was expressed as a percentage of the nominal value. Criteria of acceptance are: 66% of the total QCs must be $100 \pm 15\%$ nominal values, and 1 out of 2 QCs per level must be $100 \pm 15\%$ nominal values. Mice experiments and analysis were performed by Pharmacelsus GmbH. The pharmacokinetic parameters were calculated on the mean of the H27 concentrations at each time point. The area under the curve (AUC) was determined by a mixed log-linear method. The half-life of compound H27 $T_{1/2}$ was obtained using the following equation: $T_{1/2} = Ln\ 2/$elimination constant rate.

## Body weight and clinical score analysis

Twelve humanised mice were treated intraperitoneally every 3–4 days from D37 to D44 and every 2 days from D48 to D69 with the vehicle ($n = 6$) or with H27 at 3 mg / kg ($n = 6$). The vehicle was 30% of Kolliphor HS15/H20 (70%). The stock solution for H27 included 14 mg of H27 compound dissolved in 30% of Kolliphor HS15/H20 (70%) at a final

**The paper explained**

**Problem**

A major unresolved challenge to curing HIV-1 is that it persists in the nucleus of infected cells. Commonly administered antiretroviral drugs do not prevent nuclear entry of HIV-1 and no specific inhibitor of HIV-1 nuclear entry has to this day been identified.

**Results**

To identify compounds that prevent HIV-1 capsids from engaging with the nuclear import machinery, we screened for molecules that could reduce binding of Transportin-1 to CA hexamers. We identified one main hit, H27, and a series of structural analogues that were strongly antiviral and specifically blocked the nuclear import step. Unlike other CA-targeting compounds such as Lenacapavir (LEN, GS-6207) or PF74, this new family of inhibitors does not affect CA assembly or disassembly and retained antiviral activity against mutants that were no longer sensitive to PF74/LEN. H27 exhibited good metabolic stability in vivo and was efficient against different HIV-1 subtypes and circulating recombinant forms from treatment-naïve patients as well as HIV-1 strains resistant to the four main classes of antiretroviral drugs.

**Impact**

This work introduces compounds that specifically inhibit HIV-1 nuclear import. The pharmacological inhibition of viral nuclear entry could be an interesting new target for all viruses that replicate in the nucleus.

concentration of 2 mM (1.275 mg/mL). Mice were monitored daily for unexpected signs of distress. Body weight was measured once a week. The weight loss was calculated in reference to the weight at D0. Clinical scores were obtained by scoring multiple clinical signs (coat, movement, activity, paleness, body weight). The ethical limit was set at a total clinical score of >7.

## Statistical analyses

All statistical analyses were performed using GraphPad Prism 8. Exact *p* values are provided in Appendix Table S1. Technical replicates, biological replicates, and independent experiments are indicated in the Figure legends.

# Data availability

The 3D coordinates of HIV-1 CA hexamer in complex with TRN-1 used for virtual screening and the simulation trajectory with H27 from this publication have been deposited to the Zenodo database and assigned the identifier (https://doi.org/10.5281/zenodo.11060905).

The source data of this paper are collected in the following database record: biostudies:S-SCDT-10_1038-S44321-024-00143-w.

# Peer review information

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

## Acknowledgements

We thank Yves L. Janin (Muséum National d'Histoire Naturelle, Paris, France) for helpful discussions on medicinal chemistry during the preparation of the manuscript and Giang Ngo for help with the SPR analysis. SPR experiments were carried out using the facilities of the Montpellier Proteomics Platform (PPM-PP2I, BioCampus Montpellier). This work was funded by grants from Sidaction (AAP31-1-AEQ-12643), ANRS-MIE (ECTZ209411), Université de Montpellier (iSite) and an intramural grant (BIOLuM 2021). The work was supported by the Francis Crick Institute, which receives its core funding from Cancer Research UK (CC2029), the UK Medical Research Council (CC2029) and the Wellcome Trust (CC2029). NMR spectra were recorded at the MRC UK Biomedical NMR Facility, Francis Crick Institute, which is funded by Cancer Research UK (CC1078), the UK Medical Research Council (CC1078), and the Wellcome Trust (CC1078). We thank Elodie Jublanc (Montpellier Resources Imagerie) and Jim Zoladek (IRIM, Montpellier) for help with microscopy, Antoine Gross (IRIM, Montpellier) for sharing reagents and for helpful discussions, and Chris Aiken (Vanderbilt University) for sharing CA mutants.

## Author contributions

**Aude Boulay**: Formal analysis; Validation; Investigation.
**Emmanuel Quevarec**: Formal analysis; Validation; Investigation; Visualisation.
**Isabelle Malet**: Resources; Investigation. **Giuseppe Nicastro**: Resources; Investigation; Visualisation. **Célia Chamontin**: Investigation. **Suzon Perrin**: Investigation. **Corinne Henriquet**: Formal analysis; Investigation. **Martine Pugnière**: Data curation; Formal analysis; Supervision; Validation; Visualisation; Methodology; Writing—original draft; Writing—review and editing. **Valérie Courgnaud**: Resources; Funding acquisition; Visualisation. **Mickaël Blaise**: Conceptualisation; Formal analysis; Funding acquisition; Investigation; Writing—original draft; Writing—review and editing. **Anne-Geneviève Marcelin**: Resources; Supervision. **Ian A Taylor**: Resources; Data curation; Software; Formal analysis; Supervision; Funding acquisition; Validation; Visualisation; Methodology; Writing—original draft; Writing—review and editing. **Laurent Chaloin**: Conceptualisation; Data curation; Software; Formal analysis; Validation; Investigation; Visualisation; Methodology; Writing—original draft; Writing—review and editing. **Nathalie J Arhel**: Conceptualisation; Formal analysis; Supervision; Funding acquisition; Validation; Investigation; Visualisation; Writing—original draft; Project administration; Writing—review and editing.

Source data underlying figure panels in this paper may have individual authorship assigned. Where available, figure panel/source data authorship is listed in the following database record: biostudies:S-SCDT-10_1038-S44321-024-00143-w.

## Disclosure and competing interests statement

The authors declare no competing interests.

# Expanded View Figures

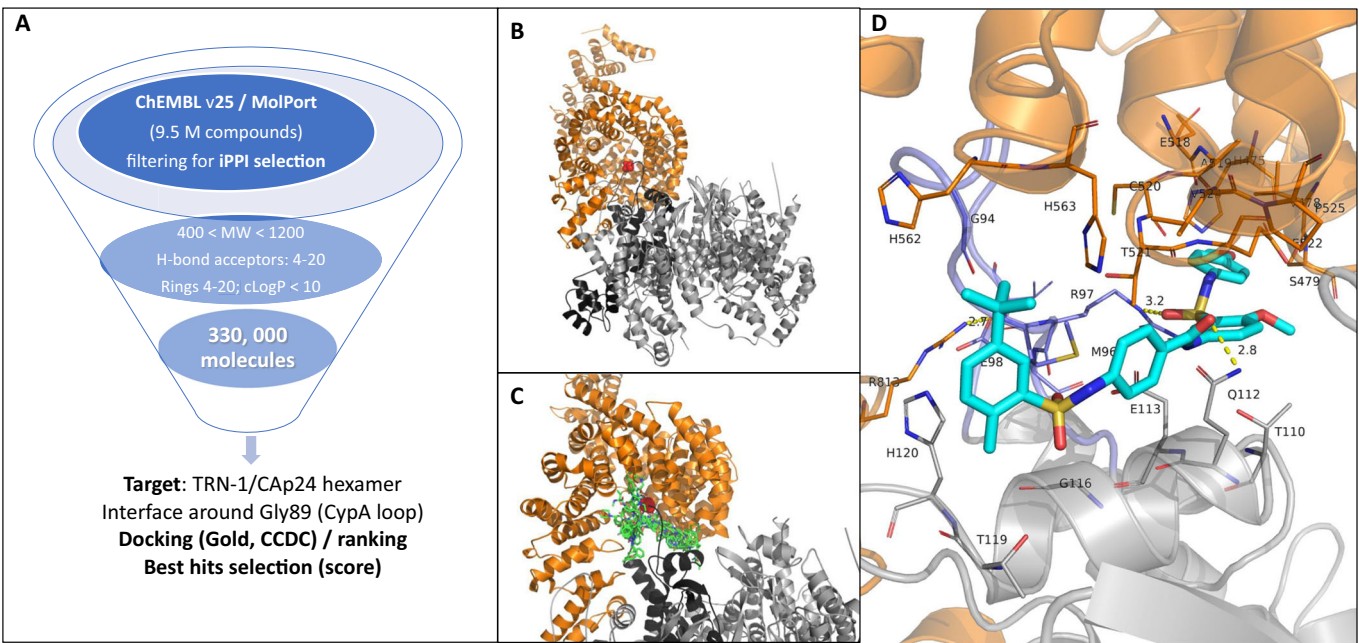

**Figure EV1.  Workflow for discovering new inhibitors of TRN-1 / CA hexamer interface.**

(A) Processing of the chEMBL (release 25, 1.9 M molecules) and MolPort (7.6 M of compounds) databases by applying specific criteria to enrich the final library in iPPI (higher molecular weight, lower solubility and high number of hydrogen bond acceptors or rings) resulting in 330,00 final compounds. (B) TRN-1/CA hexamer complex obtained by rigid docking. TRN-1 is depicted in orange cartoon representation and CA in grey; Gly89 from the CypA loop is shown as red spheres. (C) Illustration of the virtual screening using iPPI compounds with the superimposition of 10 first best-ranking hits obtained by docking with the Gold programme (green sticks). (D) Zoom-in view of the interface with residues from TRN-1 or CA interacting with the hit H27 (depicted as cyan sticks).

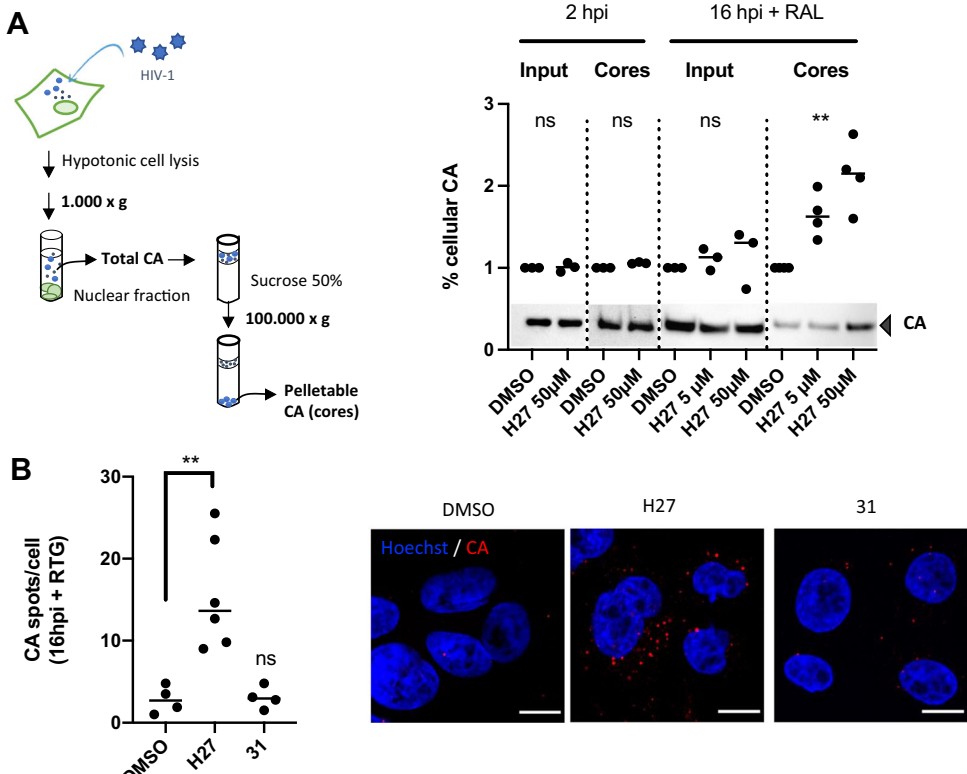

**Figure EV2. CA accumulation in cells treated with H27.**

(**A**) Fate-of-capsid assay. HeLa-R5 cells were treated with 5 or 50 μM H27 for 2 h, then inoculated with LAI-VSV-G at 1 ng p24/1000 cells for 4 h in serum-free medium to promote cell attachment, and incubated in 10% FCS for a further 2 or 16 h to allow infection to proceed. For the latter time point, cells were treated additionally with RAL (10 μM) to prevent de novo gag synthesis. Cells were collected and lysed in hypotonic buffer, and lysates were either directly analysed by immunoblotting (Input), or were centrifuged through a sucrose cushion to separate pelletable CA (cores) from soluble CA. Results show the immunoblot quantifications from three independent experiments, normalised for the DMSO control. A representative blot is provided below. Statistical significance was assessed by paired non-parametric *t*-test at 2 hpi (Wilcoxon test) and One-way ANOVA at 16 hpi (Friedman test). **$p \leq 0.01$, ns non-significant. (**B**) HeLa-R5 cells were treated with 50 μM H27 or compound **31**, then infected with LAI-VSV-G at 1 ng p24/1000 cells. The effect of compounds on HIV-1 CA accumulation was assessed at 16 hpi with RAL to prevent de novo Gag synthesis. Cells were labelled with anti-CA antibody, stained with Hoechst, and imaged using an LSM880 confocal microscope with 63 x objective. Analysis was performed using the spot detection tool in Imaris 9. The graph indicates the average number of spots per cell for 4 independent experiments and a total of >400 cells. Statistical significance was assessed by ordinary one-way ANOVA with multiple comparison. **$p \leq 0.01$, ns non-significant. Representative confocal images show CA and Hoechst staining. Scale bar = 10 μm.

**A**

"o, m, p": ortho, meta, para

**B**

Metabolism alerts:
- 4-MeOaniline
- Quinone-prone metabolites

- F (or Cl) in ortho and/or para are possible (although associated with some cytotox.).
- Larger O-*n*propyl (or the even bigger sulfamides of H27) are also possible but do not provide significant improvements.

An H, instead of MeO, appears to be possible.

**Figure EV3. Structure activity relationship (SAR).**

(A) Chemical structures of the H27 analogues 41 to 85. (B) Conclusions from the structure-activity relationship study.

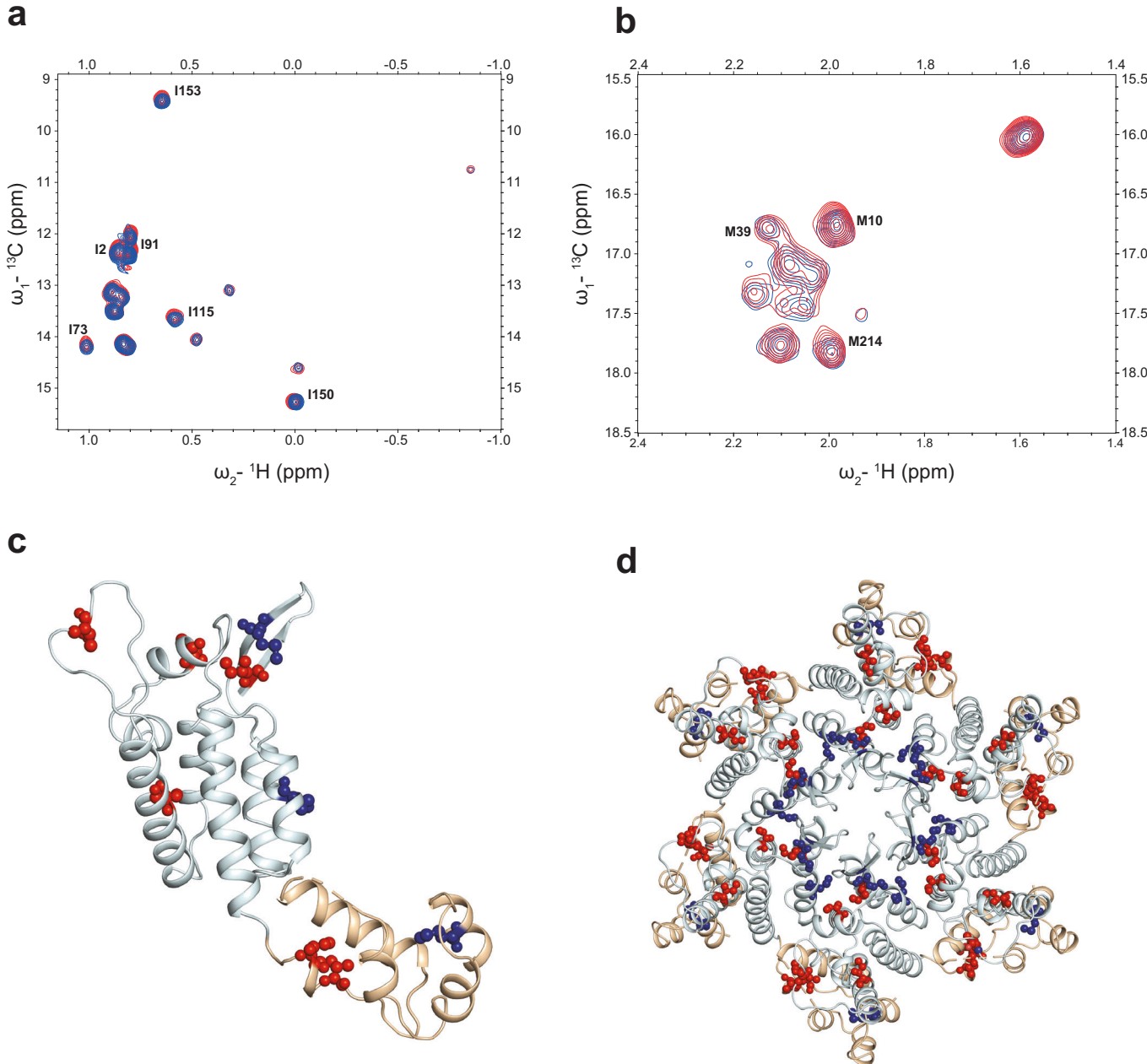

**Figure EV4.** ¹H-¹³C methyl-TROSY NMR mapping of CA-Hex H27 interactions.

(A, B) ¹H-¹³C HMQC methyl-TROSY spectra of 50 μM ¹³C Ile-δ1 - Met-ε labelled CA-hexamer (blue) and upon addition of 1 mM H27 (red). The Ile region of the spectrum is shown in (A) and the Met region in (B). Introduction of H27 results in only very small chemical shift perturbation <0.1 ppm of the assigned resonances, indicated. (C, D) Crystal structure of a CA-monomer (C) and CA-hexamer (D) taken from, PDB ID: 7ZUD. The protein backbone is shown in cartoon representation, CA-NTD is coloured cyan and CA-CTD is coloured wheat. Ile and Met residues displaying chemical shift perturbation are shown in stick and ball representation mapped onto the CA-monomer and hexamer structures, Ile in red and Met in blue. Residues that do show small chemical shift perturbations are largely dispersed across the structure.

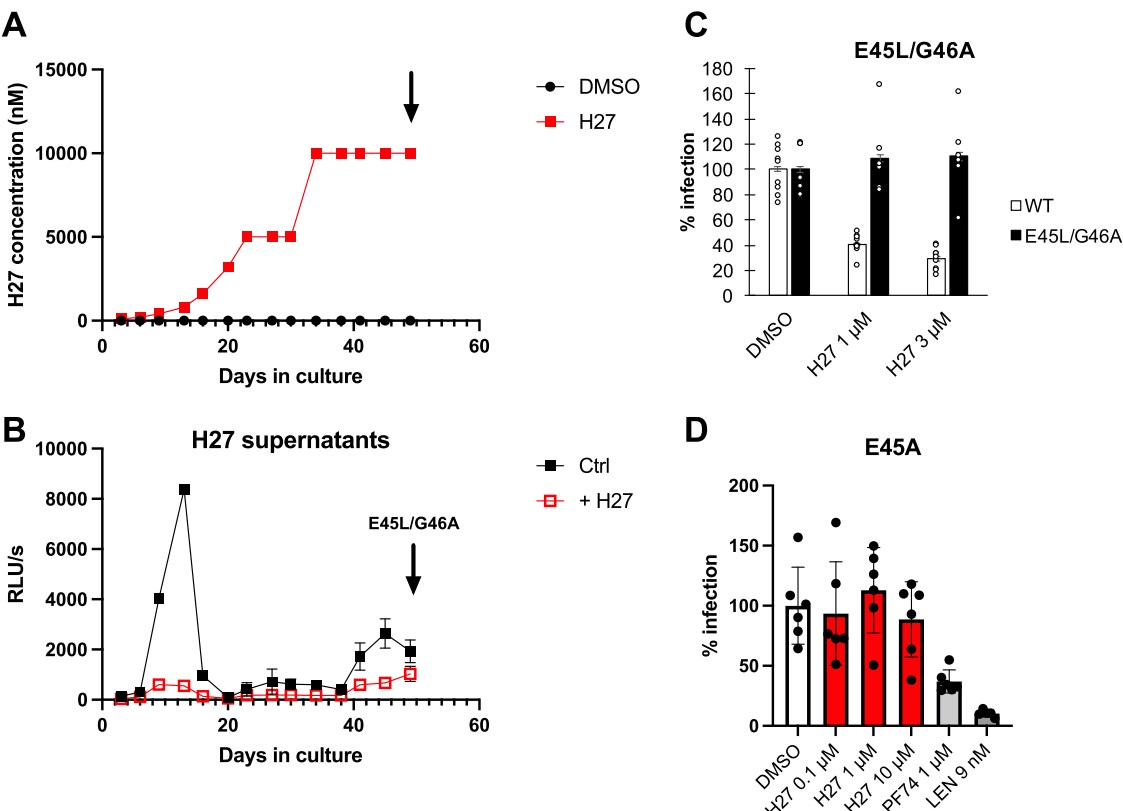

**Figure EV5. Capsid mutants that escape H27.**

(A) JLTR-R5 cells were inoculated with NL4-3 replicative virus at MOI 0.1 in the presence of 100 nM H27. Viral replication was assessed by tat transactivation of the LTR-eGFP reporter and H27 was increased by increments of 2–3-fold starting with 0.2 μM on day 1, until 10 μM from day 34 onwards. Control infections were performed in the presence of DMSO and NVP. (B) Virus production and sensitivity to H27 (10 μM) was assessed by inoculating HeLa-R5 cells with 25 μl supernatants from JLTR-R5 cultures, which is ~1:80th of the supernatant volume. β-galactosidase activity indicative of single-cycle infection was assessed at 48 hpi. Results are the mean of 3 independent experiments ± SD. The supernatants at D49 (arrow) exhibited some resistance to H27. TOPO-cloning and sequencing of capsid sequences uncovered a double E45L/G46A escape mutation. (C) Site-directed mutagenesis was performed to introduce the E45L/G46A mutations in pNL4-3. Sensitivity to H27 was tested by comparing it with wild-type virus in HeLa-R5 cells. Results show individual and mean values from three independent experiments ± SEM. (D) Sensitivity of the E45A mutant to H27. HeLa-R5 cells were treated with 0.1, 1 or 10 μM H27, 1 μM PF74 or 9 nM LEN, then infected with the E45A R9 virus. Results show the mean normalised infectivity values ± SD obtained at 48 hpi from three independent experiments.

