## [Peer Review File · EMBO Molecular Medicine]

A new class of capsid-targeting inhibitors that specifically block HIV-1 nuclear import

Aude Boulay, Emmanuel Quevarec, Isabelle Malet, Giuseppe Nicastro, Célia Chamontin, Suzon Perrin, Corinne Henriquet, Martine Pugnère, Valerie Cournaud, Mickaël Blaise, Anne-Geneviève Marcelin, Ian Taylor, Laurent Chaloin, and Nathalie Arhel

Corresponding author: Nathalie Arhel (nathalie.arhel@irim.cnrs.fr)

Review Timeline:

Submission Date:	5th Mar 24
Editorial Decision:	19th Apr 24
Revision Received:	16th Jun 24
Editorial Decision:	15th Aug 24
Revision Received:	29th Aug 24
Accepted:	2nd Sep 24

Editor: Zeljko Durdevic

Transaction Report:

19th Apr 2024

Dear Dr. Arhel,

Thank you for the submission of your manuscript to EMBO Molecular Medicine, and please accept my apologies for the delay in getting back to you, which is due to the fact that one referee needed more time to complete his/her review. We have now received feedback from the three reviewers who agreed to evaluate your manuscript.

As you will see from the reports pasted below, while the referees #1 and #3 are overall supportive raising important but minor concerns, referee #2 raises serious concerns particularly regarding the inconclusiveness and inconsistency of the results and premature nature of the study. After the cross-commenting discussion it became clear that the focus of the major revision should be on providing more mechanistic insight and providing more details about the experimental methodologies and study design. Additional experiments that further strengthen the main conclusions of the study would be appreciated. If you would like to discuss further the points raised by the referees, I am available to do so via email or video. Let me know if you are interested in this option.

We would welcome the submission of a revised version within three months for further consideration. Please let us know if you require longer to complete the revision.

Please use this link to login to the manuscript system and submit your revision: <https://embomolmed.msubmit.net/cgi-bin/main.plex>

I look forward to receiving your revised manuscript.

Yours sincerely,

Zeljko Durdevic

We require:

- 1) A .docx formatted version of the manuscript text (including legends for main figures, EV figures and tables). Please make sure that the changes are highlighted to be clearly visible.
- 2) Individual production quality figure files as .eps, .tif, .jpg (one file per figure). For guidance, download the 'Figure Guide PDF': (<https://www.embopress.org/page/journal/17574684/authorguide#figureformat>).
- 3) A .docx formatted letter INCLUDING the reviewers' reports and your detailed point-by-point responses to their comments. As part of the EMBO Press transparent editorial process, the point-by-point response is part of the Review Process File (RPF), which will be published alongside your paper.
- 4) A complete author checklist, which you can download from our author guidelines (<https://www.embopress.org/page/journal/17574684/authorguide#submissionofrevisions>). Please insert information in the

checklist that is also reflected in the manuscript. The completed author checklist will also be part of the RPF.

6) It is mandatory to include a 'Data Availability' section after the Materials and Methods. Before submitting your revision, primary datasets produced in this study need to be deposited in an appropriate public database, and the accession numbers and database listed under 'Data Availability'. Please remember to provide a reviewer password if the datasets are not yet public (see <https://www.embopress.org/page/journal/17574684/authorguide#dataavailability>).

.

- the medical issue you are addressing,

- the results obtained and

- their clinical impact.

13) Author contributions: You will be asked to provide CRediT (Contributor Role Taxonomy) terms in the submission system. These replace a narrative author contribution section in the manuscript.

14) A Conflict of Interest statement should be provided in the main text.

Please also suggest a striking image or visual abstract to illustrate your article as a PNG file 550 px wide x 300-800 px high.

***** Reviewer's comments *****

Referee #1 (Remarks for Author):

Boulay et al. have conducted an interesting study that unveils a new class of HIV-1 inhibitors targeting capsid nuclear import. Through virtual screening, the authors identified 40 compounds that potentially interact with the interface between HIV CA hexamers and the nuclear pore complex. Among these candidates, H27 emerged as a potent inhibitor of HIV infection across various models, including PBL. Notably, H27 demonstrated an IC₅₀ in the micromolar range with a CC₅₀ > 100 μM, indicating a favorable cytotoxic profile.

The authors showed that H27 specifically blocks nuclear import without disrupting reverse transcription, leading to cytoplasmic capsid accumulation. Although the precise mode of action of H27 remains to be fully elucidated, the data suggest that the compound establishes multiple interactions with CA hexamers and reduces CA-TRN-1 interaction without affecting CA assembly or disassembly kinetics.

H27 exhibited selectivity against HIV-1 strains (but not HIV-2 or SIV) while retaining efficacy against variants resistant to other CA inhibitors, demonstrating promising potential as a HIV-1 antiviral agent. The manuscript also provides insights into H27 inhibitory activity against diverse HIV-1 subtypes, including treatment-naïve strains and multi-drug-resistant mutants. Additionally, the assessment of in vivo cytotoxicity and compound stability in mice enhances the translational relevance of these findings.

Overall, this manuscript is well written, employing robust methodologies, comprehensive analyses, and appropriate controls. The novel mechanism of action of H27 as an inhibitor of CA nuclear import underscores its potential as a lead candidate for the development of new antiviral strategies against HIV.

Comments to be addressed by the authors:

P5, lines 143-148: The authors compare the IC₅₀ of H27 with those of other established HIV CA inhibitors, PF74 and LEN. To provide readers with a more comprehensive understanding of the cytotoxicity profile of H27 in comparison to these inhibitors, it would be valuable to include a comparison of the CC₅₀ values in Fig. EV3.

Line 145: please, change 'the low micromolar range' to specific numerical values.

P5, Fig 2A and 2B: The decision to quantify reverse transcript products at 24 hpi is somewhat surprising. The literature suggests that the peak of reverse transcription typically occurs around 6-8 hpi, as confirmed by the authors in Figure 2 E. Therefore, assessing the impact of H27 on reverse transcription within this critical time window (6-8 hpi) would provide a more convincing assessment of H27 effect on this process compared to measurements at 24 hpi.

Raltgravir is mentioned as 'RAL' at line 169 and as 'RTG' at line 171. Please, homogenize the nomenclature

Regarding Figure EV4A, it would be beneficial to ensure that the level of pelleted capsid is comparable between H27-treated and DMSO control samples at 2 hpi. This control would help to mitigate any potential bias in viral entry.

P6, lines 188-190, Fig EV5.

-The observation that the majority of CA signal clusters at the plasma membrane as early as 6 hpi in the DMSO control is

surprising. This timing likely corresponds to the peak of reverse transcription and integration (Figure 2E). Could the authors comment on that? Could the authors provide clarification on whether the anti-CA antibody used in this experiment recognizes CA monomers or hexamers? It would be helpful for readers to understand the specificity of the antibody and how it may influence the interpretation of the results.

-PF74 and LEN are known to inhibit multiple steps of HIV-1 replication, including uncoating and nuclear import. The authors confirm that, similar to H27, the CA signal remains in the cytoplasm at 2 hpi for these controls. However, PF74 and LEN also target late steps of replication such as CA assembly. Yet, the authors conclude that, contrary to what is observed for H27, the CA signal still accumulates at the cell periphery at 6 hpi in the presence of PF74 and LEN treatment. Was this observation expected? How do the authors explain these findings?

P9, lines 266 -270, Figures 3A and 3B: The authors may consider shortening the HIV-1 loop and testing whether this mutant has lost sensitivity to H27. This approach would help confirm the authors' hypothesis regarding the importance of the loop.

P10, line 315. Authors referred to PF74 in the text, whereas PF-74 is used in Figure EV11D. Please homogenize the nomenclature throughout both the text and figures.

P11, lines 353-357, Figure 5E. The authors note that PF74 failed to displace CA-CPSF6 interaction at 1 μ M, but argue that according to the literature the interaction is displaced at higher concentrations. It raises the question whether a similar concentration-dependent effect could be observed for H27 ?

Fig 6A The antiviral effect of H27 appears to be comparable to other drugs (3TC and NVP) at concentrations closer to 50 μ M rather than the low micromolar range. Could the authors further comment on this concentration-dependent efficacy? Additionally, while the authors demonstrate no in vivo toxicity in mice at blood concentrations of 2.8-5 μ M 5 minutes after injection (Fig EV13), the observed efficacy of the drug at this dose appears to be modest depending on HIV-1 isolates (Figure 6). Please, discuss this point.

P15, line 474, please define "SAR"

Referee #2 (Comments on Novelty/Model System for Author):

Mice/rats are not infected by HIV. There should be no animal studies at this early stage. They are clearly not leads and not appropriate for animal level experiments.

Referee #2 (Remarks for Author):

The manuscript by Boulay and colleagues is an extensive multidisciplinary effort that utilizes multiple techniques in an effort to describe/characterize the properties of H27, a compound proposed to have antiviral properties through blocking HIV-1 nuclear import. There is an in silico search for small molecules that may bind at the interface of CA with TRN-1 apparently based on a previous publication. The details of the docking, the starting structure and assumptions are not clear. Neither are the results are conclusive or consistent with biophysical or resistance data presented here. The H27 hit is extensively characterized with virological, biophysical and others methods, including animal model studies, which in my opinion are premature, as this is far from a compound lead.

Despite the extensive efforts by the research team, there does not seem to be a specific mechanism by which the compound works. The authors themselves admit pleiotropic effects and this is clearly consistent with the lack of significant improvement through the SAR (structure activity relationship) chemistry efforts.

Given the lack of specific interactions with CA or TRN-1 it may be that H27 also interacts with host factors. It is also unclear whether H27 also affects "cellular" entry (not nuclear entry).

Given the mechanistic uncertainty it is premature for publication as a compound that "inhibits specifically nuclear entry".

Specific comments:

-Not possible to evaluate the in silico studies and searches, as there is complete information available-figures, specific coordinates of complex, details of 3D model of the CA complex with karyopherin used in the in silico studies.

-It is stated that the compound does not bind CA by itself, or TRN-1 by itself, but only the complex. A Kd of 0.34 mM is mentioned, which is very high and rather not relevant to the ~100 fold presented EC50 values. Yet, the MD simulation/figure shows binding of H27 to CA by itself. Where is the TRN-1 in this figure? What are the interactions that would be consistent with the claim?

-How come resistance appears so far at residue 45 and not where the specific docking interactions appear?
-Same with the NMR data. Why only CA and not complex with TRN-1 peptide? The design of the NMR experiment is not consistent with the proposed model of binding the complex of CA with TRN-1.
-Why are the interactions in the NMR experiments different than the resistance mutations? The following reference to the NMR data: "Residues that do show small chemical shift perturbations are largely dispersed across the structure" is a potential cause of concern for unspecific pleiomorphic interactions and not consistent with the proposed model of binding the complex of CA with TRN-1.
-Figure 3 D,E,F,G shows interactions none of which are consistent with the resistance data. Moreover, why is the simulation done with Ca hexamer when it is shown not to bind in the biophysical studies?
-Figure 2. MOIs are not mentioned. MOI for ALL experiments should be provided.
-"CANC and TRN-1 expression and purification. The purification of HIV-1 CANC and TRN-1 was described in a previous publication (Fernandez et al., 2019)." More details at least for TRN-1 that is not a standard protocol should be provided.
-The background of CA spots intranuclearly is very high. Why? Is the MOI realistic?
-It is stated that the potency of LEN observed here is similar to that reported in other papers. To our knowledge, instead of nM observed here, the potency for LEN has been reported to be picomolar.
-The data in Figure 1 show a difference in inhibition of viruses that have different modes of cellular (not nuclear) entry. Depending on whether the entry is through VSV, CXCR4 or CCR5-based systems, there are significant differences: at 10 uM H27 there is ~90% inhibition of the VSV-based system, whereas barely any inhibition for the other two. This is consistent with the potential multiple modes of binding of a compound that in terms of medicinal chemistry may be binding in multiple modes and multiple targets -essentially mentioned by the authors as well.
-The failure of the SAR to result in any significant improvement in potency is consistent with multiple effects.
-"Using a 3D model of HIV-1 CA hexamers bound by the karyopherin TRN-1 based on our previous mutant studies (Fernandez et al., 2019), we searched chemical databases for compounds that might bind at this interface." Lack of detail makes it impossible to understand and evaluate the strategy. Figure of complex? Docking site? Interactions? The MD experiment are rather not helpful.
The statement "To discover potential HIV-1 nuclear import inhibitors, we performed in silico screening on the CA 117 hexamer - TRN-1 complex, using a 3D model based on previous mutagenesis studies (Fernandez et al., 2019) and the available structural information for TRN-1 and CA" does not really help.
-The CA hexamer - TRN-1 complex based on two crystal structures (4FQ3 for TRN-1 and 3H4E for CA protein) was modelled as previously described (Fernandez et al., 2019). Also does not help.
-"A total of 330,000 iPPI-like compounds were screened at the HIV-1 CA hexamer - TRN-1 interface using 20 genetic algorithms for the search of docking solutions..." What are the coordinates of complex? How can we evaluate these protocols?

"We screened for compounds that are likely to bind the HIV-1 CA hexamer - TRN-1 interface. This allowed the identification of a new family of inhibitors that specifically block HIV-1 nuclear import." It is not possible to understand what was done here and why the results are incongruent with what appears to be a different strategy.
-Disassembly assay is not really a good representation of what is happening in the cellular context; The standard is live microscopy-for relevant assays Pathak, Melikian, Hope, and other labs.
-"Although we previously showed that in HIV-1 the glycine at position 89 is optimally positioned for binding to TRN-1 (Fernandez et al., 2019), G89V mutants were still sensitive to H27, confirming that H27 makes multiple contact points with the CA". The resistance experiments do not seem to confirm or be consistent with the in silico study.

Referee #3 (Remarks for Author):

This manuscript describes the identification of H27 and structural analogues that were found to bind HIV-1 capsid (CA) and prevent import through the nuclear pore. The discovery of H27 is predicated upon the use of in silico approaches, but significant biological data is presented to validate the identification of this hit compound. An SAR study was also done using commercially available analogues of H27 to identify 5 additional active compounds that do not show significant cytotoxicity. This portion of the manuscript is somewhat limited as these compounds do not appear to be considered as viable leads for development since nearly all of the work was done with H27. In addition, one limitation of this SAR work was in the limited variation of the "left" side of the molecules as shown in Figure EV6. The data obtained suggests that the right-hand of the molecule is not critical for activity, suggesting that key contacts are made in the other three rings. While the third ring is varied through the introduction of various substituents and a variety of substitution patterns, the first and second rings (from the left) are largely held constant, limiting the overall value of the conclusions that can be drawn. This is a relatively minor concern as this is not the focus of the paper. Overall, the paper is well constructed and identifies a molecule that acts via a novel mechanism. This could provide a nice starting point for development of a class of compounds that act at this particular site.

There are some minor issues that should be addressed, however, if this paper is accepted for publication:

Line 35-36: Add the word "as" in the phrase "as well as HIV-1 strains"

Line 37: Stating that this is a new class of antivirals suggests that these are approved therapeutics. It would be more appropriate to say that these compounds demonstrate a novel mechanism of action.

Line 42: Recommend either "by active transport" or "by an active transport mechanism".

Line 129 (and after): It is recommended that bold font be used when indicating compound numbers.

Line 132: The molecular formula should be presented using subscripted numbers.

Table EV1: Compounds 8, 9, 17, and 24 are listed as NOT AVAILABLE. This would seem to indicate that those compounds could not be obtained for subsequent screening. If this is the case, why are they included in the list of compounds 1-40, which is stated to have been screened (Line 129).

Line 223-226: It is not clear here what the authors are suggesting about the SAR here, particularly with respect to fluorination. Compounds 55, 49, and 84 are among the most active and show limited cytotoxicity. Compounds 80 and 81 are also fluorinated (in similar positions) and show significant cytotoxicity.

Figure EV6: The order of the numbering of the compounds in part A is a bit confusing. If the authors wish to group compounds that share similar substitution patterns (e.g. an ortho, meta, and para series), consider renumbering so that these numbers are in sequential order. In the current arrangement, it becomes difficult to find certain numbers (e.g. compound 49, which follows 55 and 70 in the list). Also, the labels in part B should be reviewed. In particular, the "H also possible" is not clear since the arrow points to the methyl group of the methoxy. I believe that the "H" would refer to complete removal of that methoxy group, rather than simply the presence of a phenol. In addition, if using fluorine, chlorine should also be used. Alternatively, fluoride/chloride could be used, but since the reference is to the atom itself (rather than the molecule), fluorine/chlorine would likely be more appropriate. Finally, the meaning of "isopropyl of large sulfonamides" is not clear, specifically what is meant by a "large sulfonamide".

Line 227-228: The statement that the mechanism of action of the analogues is linked to their structure is the very definition of the term "structure-activity relationship". This statement, therefore, is a bit redundant and not necessary. The value of this sentence, however, is the recognition of the high degree of structural specificity required for activity.

Montpellier, 16th June 2024

Dear Dr. Durdevic,

We thank you for the feedback on our manuscript entitled “Identification of a new class of capsid-targeting inhibitors that specifically block HIV-1 nuclear import”.

The reviewers' comments were very helpful and we wish to thank them for the time they committed to reviewing our work and for their appreciation of the study.

We identified 4 main concerns, which were addressed as follows:

1. Insufficient protocol descriptions: we now provide extensive detail of the docking simulations in the M&M section and **Fig. EV1 new**, as well as a detailed protocol for the purification of TRN-1. We also provide IC₅₀, CC₅₀ and MOI values, as requested. The 3D coordinates of HIV-1 CA hexamer in complex with TRN-1 used for virtual screening and the simulation trajectory with H27 have also been deposited to the Zenodo database.
2. Specificity of H27: we provide fate-of-capsid assays at 2 hpi, as requested, to demonstrate that H27 does not affect cellular entry (**Fig. EV6 new**) and we compared envelope tropism for sensitivity to H27 (**Fig. EV4A new**). We also tested the effect of H27 on 10 well-characterized shuttling proteins and found no effect (**Fig. EV8 new**).
3. Specificity of the capsid antibody: we tested the binding affinity of the AG3.0 antibody to CA monomers and hexamers by SPR and provide K_D values (**Fig. EV5 new**).
4. The timing of infection experiments: we have repeated the quantification of reverse transcription at 7 hpi, as requested (**Fig. EV4D new**), we also clarify in the M&M section how we define hpi values

New or edited figure panels or tables are indicated in bold in our point-by-point response below.

Thank you for considering our manuscript and we look forward to hearing from you.

Best regards,

Nathalie Arhel

***** Reviewer's comments *****

Referee #1 (Remarks for Author):

Boulay et al. have conducted an interesting study that unveils a new class of HIV-1 inhibitors targeting capsid nuclear import. Through virtual screening, the authors identified 40 compounds that potentially interact with the interface between HIV CA hexamers and the nuclear pore complex. Among these candidates, H27 emerged as a potent inhibitor of HIV infection across various models, including PBL. Notably, H27 demonstrated an IC₅₀ in the micromolar range with a CC₅₀ > 100 μM, indicating a favorable cytotoxic profile.

The authors showed that H27 specifically blocks nuclear import without disrupting reverse transcription, leading to cytoplasmic capsid accumulation. Although the precise mode of action of H27 remains to be fully elucidated, the data suggest that the compound establishes multiple interactions with CA hexamers and reduces CA-TRN-1 interaction without affecting CA assembly or disassembly kinetics.

H27 exhibited selectivity against HIV-1 strains (but not HIV-2 or SIV) while retaining efficacy against variants resistant to other CA inhibitors, demonstrating promising potential as a HIV-1 antiviral agent. The manuscript also provides insights into H27 inhibitory activity against diverse HIV-1 subtypes, including treatment-naïve strains and multi-drug-resistant mutants. Additionally, the assessment of in vivo cytotoxicity and compound stability in mice enhances the translational relevance of these findings.

Overall, this manuscript is well written, employing robust methodologies, comprehensive analyses, and appropriate controls. The novel mechanism of action of H27 as an inhibitor of CA nuclear import underscores its potential as a lead candidate for the development of new antiviral strategies against HIV.

Comments to be addressed by the authors:

P5, lines 143-148: The authors compare the IC₅₀ of H27 with those of other established HIV CA inhibitors, PF74 and LEN. To provide readers with a more comprehensive understanding of the cytotoxicity profile of H27 in comparison to these inhibitors, it would be valuable to include a comparison of the CC₅₀ values in Fig. EV3.

Response: We thank the reviewer for this suggestion. CC₅₀ values for PF-74 and LEN have been added to **Table EV2 new**.

Line 145: please, change 'the low micromolar range' to specific numerical values.

Response: The specific numerical IC₅₀ values have been added to the manuscript (line 146).

P5, Fig 2A and 2B: The decision to quantify reverse transcript products at 24 hpi is somewhat surprising. The literature suggests that the peak of reverse transcription typically occurs around 6-8 hpi, as confirmed by the authors in Figure 2 E. Therefore, assessing the impact of H27 on reverse transcription within this critical time window (6-8 hpi) would provide a more convincing assessment of H27 effect on this process compared to measurements at 24 hpi.

Response: We thank the reviewer for pointing out that late reverse transcripts (LRT) peak around 6-8 hpi. In our manuscript, we chose to quantify LRT at 24 hpi, because we wanted to use the same DNA extracts that are used for the quantification of 2-LTR circles, which peak in abundance at 24 hpi. However, to address this reviewer's concern, we repeated the LRT experiments at 7 hpi and now include the quantification of LRT at 7 hpi in **Fig. EV4D new**.

Raltegravir is mentioned as 'RAL' at line 169 and as 'RTG' at line 171. Please, homogenize the nomenclature

Response: Raltegravir has been homogenized to RAL throughout.

Regarding Figure EV4A, it would be beneficial to ensure that the level of pelleted capsid is comparable between H27-treated and DMSO control samples at 2 hpi. This control would help to mitigate any potential bias in viral entry.

Response: We agree that this is an important control. In Fig. EV4A (now Fig. EV6A), "Input" was already provided and corresponds to the amount of CA that was harvested from infected cells at the indicated time points. This control already suggested that there was no difference in viral entry between samples. However, to fully address this reviewer's concern, we measured total CA and pelleted CA at 2 hpi as requested, and now include these data in a revised **Fig. EV6A new**.

P6, lines 188-190, Fig EV5.

-The observation that the majority of CA signal clusters at the plasma membrane as early as 6 hpi in the DMSO control is surprising. This timing likely corresponds to the peak of reverse transcription and integration (Figure 2E). Could the authors comment on that?

Response: Thank you for pointing out this discrepancy. We have clarified our M&M section and legend to more clearly state how we define times of infection (lines 882-888). All plate assays (e.g. Fig 2E) are performed in 5% FCS serum throughout, and hours post-infection (hpi) refer to the time after addition of virus. For imaging/IP/capsid pulldown assays, infection is preceded by a 2 h viral attachment step (in serum-free medium), and hpi refer to the time after addition of FCS.

Having said this, this reviewer is right in pointing out that this experiment was performed rather early. We did this to ensure that no *de novo* particles were being produced.

Could the authors provide clarification on whether the anti-CA antibody used in this experiment recognizes CA monomers or hexamers? It would be helpful for readers to understand the specificity of the antibody and how it may influence the interpretation of the results.

Response: Previous work showed that the AG3.0 monoclonal antibody has broad reactivity recognizing the Gag capsid protein (p24-27) and Gag precursors p38, p55, and p150 of HIV-1, HIV-2, SIVmac, and SIVagm (Sanders-Beer et al. Virology 2012). However, we are not aware of any study that determined whether it recognizes CA monomer or hexamer, and binding affinities.

To address this point experimentally, we performed SPR experiments using anti-mouse Fc to

immobilize AG3.0 and tested this antibody's binding affinity to CA monomers and hexamers. Results indicated that the antibody binds both to CA monomers and hexamers, with an affinity constant of $K_D = 4,12E-09$ M and $5,39E-08$ M, respectively (Fig. EV5 new).

-PF74 and LEN are known to inhibit multiple steps of HIV-1 replication, including uncoating and nuclear import. The authors confirm that, similar to H27, the CA signal remains in the cytoplasm at 2 hpi for these controls. However, PF74 and LEN also target late steps of replication such as CA assembly. Yet, the authors conclude that, contrary to what is observed for H27, the CA signal still accumulates at the cell periphery at 6 hpi in the presence of PF74 and LEN treatment. Was this observation expected? How do the authors explain these findings?

Response: Yes, this observation was expected because drugs were added prior to infection, and therefore only the first block can be observed. We did however test the drugs on the late steps of replication in Fig. 2G, and this indeed confirms that PF-74 and LEN target the late steps.

P9, lines 266 -270, Figures 3A and 3B: The authors may consider shortening the HIV-1 loop and testing whether this mutant has lost sensitivity to H27. This approach would help confirm the authors' hypothesis regarding the importance of the loop.

Response: This is a great suggestion and we have indeed considered making point mutations and deletions in the Cyp-loop to look for loss of H27 sensitivity. However, previous attempts at making point mutants G89A and P90A (Braaten et al 1996 & Braaten et al 2001), R100A/S102A (Von Schwedler et al 2003) and deletions ($\Delta 87-97$) (Ganser-Pornillos et al 2004) have a variety of effects, including on assembly, and invariably impact on virus infectivity. Given the Cyp-loop's connection with viral fitness we are concerned that interpretation of an H27 effect will be compromised as it will be difficult to discern from Cyp-loop deletion effects that may abolish infection, dominate-over or even cooperate with H27 interactions.

P10, line 315. Authors referred to PF74 in the text, whereas PF-74 is used in Figure EV11D. Please homogenize the nomenclature throughout both the text and figures.

Response: PF74 has been homogenized throughout.

P11, lines 353-357, Figure 5E. The authors note that PF74 failed to displace CA-CPSF6 interaction at 1 μ M, but argue that according to the literature the interaction is displaced at higher concentrations. It raises the question whether a similar concentration-dependent effect could be observed for H27 ?

Response: We agree that we cannot exclude that H27 could disrupt CA-CPSF6 binding at much higher concentrations, therefore we have added the following sentence to acknowledge this: "Of note, PF74 also failed to displace CPSF6 at 1 μ M, which is concordant with others who reported that this may only be achieved at high concentrations (15-25 μ M) (Francis et al., 2020, Muller et al., 2021), and therefore we cannot exclude that H27 might have a similar concentration-dependent effect on CPSF6." (line 374)

Fig 6A The antiviral effect of H27 appears to be comparable to other drugs (3TC and NVP) at concentrations closer to 50 μ M rather than the low micromolar range. Could the authors further comment on this concentration-dependent efficacy? Additionally, while the authors demonstrate no in vivo toxicity in mice at blood concentrations of 2.8-5 μ M 5 minutes after injection (Fig EV13), the observed efficacy of the drug at this dose appears to be modest depending on HIV-1 isolates (Figure 6). Please, discuss this point.

Response: The IC50 values for 3TC and NVP are ~ 0.3 μ M and ~ 0.08 μ M, respectively. In Fig. 6A, they were used at 5 μ M and 1 μ M, respectively, i.e. 17x and 12.5x their IC50 concentration. The IC50 value for H27 is 3 μ M, therefore full inhibition by H27 is comparable to 3TC and NVP at 38-51 μ M. The toxicity experiments in mice were carried out by injecting 1 mg/kg (18.5 μ M in blood i.e. ~ 35 x in vitro IC50) and 3 mg/kg (55 μ M in blood i.e. ~ 100 x in vitro IC50).

P15, line 474, please define "SAR"

Response: This is now defined line 219.

Referee #2 (Comments on Novelty/Model System for Author):

Mice/rats are not infected by HIV. There should be no animal studies at this early stage. They are clearly not leads and not appropriate for animal level experiments.

Response: We agree that H27 is a hit, not a lead. This is the reason why our work does not include lead experiments such as *in vivo* efficacy trials. However, DMPK *in vivo* assays (half-life and toxicity) are important and considered as a necessary part of hit-to-lead experiments. The protocol received ethical authorization and was performed on a very limited number of mice, thus respecting the Three Rs guiding principles for more ethical use of animals in scientific research.

Referee #2 (Remarks for Author):

The manuscript by Boulay and colleagues is an extensive multidisciplinary effort that utilizes multiple techniques in an effort to describe/characterize the properties of H27, a compound proposed to have antiviral properties through blocking HIV-1 nuclear import. There is an *in silico* search for small molecules that may bind at the interface of CA with TRN-1 apparently based on a previous publication. The details of the docking, the starting structure and assumptions are not clear. Neither are the results are conclusive or consistent with biophysical or resistance data presented here.

Response: We apologize for the lack of clarity on this part, which can be confusing since two types of simulations have been performed. The first one was just to closely analyze the interactions between TRN-1 and the CA hexamer before doing the virtual screening and most importantly to start with an energy-minimized structure. Briefly, the previously obtained 3D model of TRN1- CA complex (Fernandez et al. 2019) was subjected to an intensive conformational sampling (several heating and cooling cycles) to evaluate the stability of the complex in order to start the virtual screening on a stable, relaxed conformation. From this screening on TRN1-CA complex, H27 was identified as a hit with high potential of interference during the complex formation.

The second part of simulations was performed to study H27 binding to the CA hexamer in the absence of TRN-1 since SPR experiments showed that H27 binds the CA hexamer, albeit with weak affinity (Fig. EV3).

We performed longer simulations (500 ns) with only H27 and the CA hexamer either from HIV-1 (using the crystal structure 3H4E as starting structure) or SIVmac (using the AlphaFold model as starting structure). In these simulations, H27 was positioned 30 angstroms above the CA hexamer before starting the simulation and it was allowed to freely move around CA hexamer (no constraint was applied between protein and ligand).

We are aware that the results, like all simulations, are merely predictive, but they are consistent in one respect: the preferential binding sites selected by H27, at the vicinity of PR loop and preferentially at the interface between two adjacent CA monomers.

We apologize for this part that was indeed not detailed enough in the submitted version of the manuscript, especially in the M&M (lines 824-872) and Results (lines 285-288) sections. This has been corrected now in the revised version and we have included **Fig. EV1 new**.

The H27 hit is extensively characterized with virological, biophysical and others methods, including animal model studies, which in my opinion are premature, as this is far from a compound lead. Despite the extensive efforts by the research team, there does not seem to be a specific mechanism by which the compound works. The authors themselves admit pleiotropic effects and this is clearly consistent with the lack of significant improvement through the SAR (structure activity relationship) chemistry efforts.

Given the lack of specific interactions with CA or TRN-1 it may be that H27 also interacts with host factors. It is also unclear whether H27 also affects "cellular" entry (not nuclear entry).

Given the mechanistic uncertainty it is premature for publication as a compound that "inhibits specifically nuclear entry".

Response: We thank this reviewer for highlighting 2 potential concerns here.

First, H27 is a hit, not a lead. We absolutely agree with this and there is no claim that H27 is a lead compound in the manuscript. It is also true that the SAR study did not improve the IC50. However, this preliminary SAR does pinpoint important contribution to some chemical groups and their positioning. We agree that more molecules would be needed to draw conclusions on the role of single substituents or specific substitution patterns in terms of activity and cytotoxicity.

Second, we agree that H27 makes multiple contact points with cores and probably reduces the binding of CA to more than one cellular partner (although interestingly not CPSF6, see Fig. 5D). However, we disagree that effects are pleiotropic in the sense that H27 does block HIV-1 nuclear import specifically. The specific effects on HIV-1 nuclear import are demonstrated using multiple techniques (IF, qPCR, PLA), and looking at multiple viral components (viral DNA, integrase, CA), thus reducing potential confounding factors. Moreover, no other step of the replication cycle was found to be affected (neither cellular entry Fig. EV6A; nor reverse transcription Fig. 2A; nor late steps Fig. 2G). We have further strengthened these assays by quantifying RT at 7 hpi, instead of 24 hpi (**Fig. EV4D new**), and by performing fate-of-capsid assay at 2 hpi to control effects on cellular entry (**Fig. EV6A new**).

Furthermore, to confirm the specificity of H27, we analyzed the subcellular localization of 6 different shuttling proteins following treatment by H27: TRN-1, CPSF6, PML, IRF-3, NF- κ B, and influenza A virus nucleoprotein (NP) (**Fig. EV8 new**). We thereby show that H27 does not disrupt the nucleocytoplasmic trafficking of any tested cellular or viral shuttling protein, with the exception of HIV-1 CA. Taken together, these results advocate for H27 as a specific inhibitor of HIV-1 nuclear import.

Specific comments:

-Not possible to evaluate the *in silico* studies and searches, as there is complete information available-figures, specific coordinates of complex, details of 3D model of the CA complex with karyopherin used in the *in silico* studies.

Response: We apologize for this inconvenience that is obviously due to the lack of details of the *in silico* section, we have now added to the M&M section the detailed process of the virtual screening of chemical libraries made by docking and the next steps in performing MD simulations of the CA hexamer/H27 complex (lines 824-872). This point is described in detail in the response to the Reviewer's first point, above.

-It is stated that the compound does not bind CA by itself, or TRN-1 by itself, but only the complex.

Response: We do not state this. The compound does bind CA hexamers by itself (Fig. EV2) and we could not test binding to the CA-TRN-1 complex because H27 disassembles it (Fig. 5C).

A K_d of 0.34 mM is mentioned, which is very high and rather not relevant to the ~100 fold presented EC50 values. Yet, the MD simulation/figure shows binding of H27 to CA by itself. Where is the TRN-1 in this figure? What are the interactions that would be consistent with the claim?

Response: In SPR, H27 binds to the immobilized hexameric form of the capsid with a K_D of approximately 300 μ M without the presence of TRN1. This is in agreement with the MD results which show an interaction site between 2 CA monomers. The affinity measured in SPR is an *in-apparatus* test performed on the capsid hexamer covalently immobilized on the sensor by amine coupling. It cannot be ruled out that this K_D is different in solution (than that inferred by MD experiments) and that the IC50s obtained *in cellulo* on infected cells are much lower than the *in vitro* measured K_D.

-How come resistance appears so far at residue 45 and not where the specific docking interactions appear?

Response: The resistance mutant at residue 45 (E45L/G46A) probably changes the whole capsid morphology, or leads to structural/stability changes. Arguments in support of this are:

- We could not produce infectious virus by transfecting HEK-293T cells with the E45L/G46A construct. This could only be achieved by spiking the transfection mix with 10% wild-type NL43 (mentioned line 474 and lines 1096-1098).
- Hyperstability resulting from changes at residue 45 has been widely reported using different techniques (Forshey et al., 2002; Guedan et al., 2021; Eschbach et al., 2021, among others).

- The well-characterised hyperstable E45A mutant was also resistant to H27 (Fig. EV14D). This suggests that the mutations do not impact H27 directly and are therefore not incompatible with the contact points that were identified by NMR/divergent lentiviral capsids/simulations. Nicastro et al., 2022 used ssNMR to show how IP6 binding affects residue dynamics at sites distal to the NTD-channel. Here, IP6-binding in the NTD freezes out fixed conformations in a fluid interface 50 Å away at the CTD-trimer interface revealing allosteric communication and suggesting that mutations in one part of the shell can affect interactions and dynamics at distal sites. In this way, it is possible that E45A forces a rigidity around $\alpha 1$ - $\alpha 2$ - $\alpha 3$ that could freeze out a conformation at a distal site that is unfavorable for H27 binding.

-Same with the NMR data. Why only CA and not complex with TRN-1 peptide? The design of the NMR experiment is not consistent with the proposed model of binding the complex of CA with TRN-1.

-Why are the interactions in the NMR experiments different than the resistance mutations? The following reference to the NMR data: "Residues that do show small chemical shift perturbations are largely dispersed across the structure" is a potential cause of concern for unspecific pleiomorphic interactions and not consistent with the proposed model of binding the complex of CA with TRN-1. Response: To take the reviewer's second point first. Indeed, we only see small-dispersed effects in methyl-TROSY NMR experiments employing CA-hexamers. However, as we discuss in the text (lines 253-259), our hypothesis is that H27 effects may well be in the context of the viral shell where binding can be at an inter-subunit interface that is not present in isolated hexamers or CA monomers. This could be at inter-hexamer trimer and dimer interfaces, or to areas of a distinct curvature type across capsid shells, such as interfaces within pentamers or at vertices between pentamers and hexamers. Similarly, the resistance mutants need not map directly to a ternary or binary interface as they may well effect H27 interactions through changes to capsid stability and/or local morphology with respect to curvature. This is particularly pertinent for E45 that sits at the $\alpha 1$ - $\alpha 2$ intrahexamer interface and has precedence for capsid-stabilising effects (Forshey et al 2002) and we already discuss this point on lines 449-453.

With respect to analysing the full CA-hex-H27-TRN-1 ternary complex by NMR. We would be very happy if this were a tractable way forward. However, with the CA-hexamer alone we are already operating at a molecular weight of ~ 160 kDa and using selective IM labelling to enable the detection of H27 effects on the Ile and Met methyl's. The addition of TRN-1 (Mr = 100 kDa x6) would increase the total molecular weight even further (760 kDa) and unfortunately the accompanying spectral broadening would degrade the spectra to such an extent that it would preclude any measurements at all.

-Figure 3 D,E,F,G shows interactions none of which are consistent with the resistance data. Moreover, why is the simulation done with Ca hexamer when it is shown not to bind in the biophysical studies?

Response: Indeed, it is difficult to correlate with the resistance data since many more simulations would have been required with various combinations of mutations in order to predict accurately the impact of these mutations on H27 binding. However, we do not claim that the resistance mutations impact directly the H27 binding but rather induced a mechanism by which the stability of the capsid has changed.

The current simulations were carried out on CA hexamer because H27 showed a better affinity for the hexamer than for the monomer (as evidenced by NMR and SPR experiments) even if the affinity is unexpectedly weak, likely because H27 requires a more complex structure such as CA tube or a higher-organized CA lattice. Another assumption is that the affinity is weak because TRN-1 is not present and cannot contribute to the final affinity of this compound (making additional interactions with H27).

-Figure 2. MOIs are not mentioned. MOI for ALL experiments should be provided.

Response: We described MOIs in the M&M section: "Unless otherwise stated, infections were all carried out at 1 ng p24/1,000 cells (i.e. MOI 5, assuming that 1 ng of p24 corresponds to 5,000 transducing units (Zufferey et al., 1998))." (lines 886-888)

- "CANC and TRN-1 expression and purification. The purification of HIV-1 CANC and TRN-1 was described in a previous publication (Fernandez et al., 2019)." More details at least for TRN-1 that is not a standard protocol should be provided.

Response: The M&M section has been updated with details for the purification of TRN-1 (lines 1010-1039).

- The background of CA spots intranuclearly is very high. Why? Is the MOI realistic?

Response: Infections were carried out at MOI 5 (see M&M, lines 886-888). Although we agree that this is quite high, it is the necessary MOI to see capsid in the nucleus, as others have also reported (Selyutina et al., Cell Reports 2020; Francis et al., Viruses 2020; Scoca et al., J Mol Cell Biol. 2023).

- It is stated that the potency of LEN observed here is similar to that reported in other papers. To our knowledge, instead of nM observed here, the potency for LEN has been reported to be picomolar.

Response: We apologize for this mistake, which we have now corrected (lines 82 and 147).

- The data in Figure 1 show a difference in inhibition of viruses that have different modes of cellular (not nuclear) entry. Depending on whether the entry is through VSV, CXCR4 or CCR5-based systems, there are significant differences: at 10 μ M H27 there is ~90% inhibition of the VSV-based system, whereas barely any inhibition for the other two. This is consistent with the potential multiple modes of binding of a compound that in terms of medicinal chemistry may be binding in multiple modes and multiple targets - essentially mentioned by the authors as well.

Response: Although the degree of inhibition appeared greater with VSV-G pseudotyped viruses in Fig. 1, this only reflected the intrinsic differences in the dynamic range of the two assays (luminescence versus fluorescence). To demonstrate this, we repeated the experiments to compare the three tropisms in the same system. We confirm that all three viruses were equally sensitive to H27 when compared in the same system (HeLa-R5 cells, LTR- β gal) (**Fig. EV4A new**). Furthermore, in response to Reviewer 1, we performed additional experiments to confirm that the level of pelleted capsid is comparable in H27-treated and DMSO control samples at 2 hpi (**Fig. EV6A new**). This confirms that viral entry is unaffected by H27.

- The failure of the SAR to result in any significant improvement in potency is consistent with multiple effects.

Response: Indeed, this explanation can be true but it must be noted that the SAR study was very limited (45 compounds in total) and maybe with more compounds we would have managed to improve the potency.

- "Using a 3D model of HIV-1 CA hexamers bound by the karyopherin TRN-1 based on our previous mutant studies (Fernandez et al., 2019), we searched chemical databases for compounds that might bind at this interface." Lack of detail makes it impossible to understand and evaluate the strategy. Figure of complex? Docking site? Interactions? The MD experiment are rather not helpful.

The statement "To discover potential HIV-1 nuclear import inhibitors, we performed in silico screening on the CA 117 hexamer - TRN-1 complex, using a 3D model based on previous mutagenesis studies (Fernandez et al., 2019) and the available structural information for TRN-1 and CA" does not really help.

- The CA hexamer - TRN-1 complex based on two crystal structures (4FQ3 for TRN-1 and 3H4E for CA protein) was modelled as previously described (Fernandez et al., 2019). Also does not help.

- "A total of 330,000 iPPI-like compounds were screened at the HIV-1 CA hexamer - TRN-1 interface using 20 genetic algorithms for the search of docking solutions..." What are the coordinates of complex? How can we evaluate these protocols?

"We screened for compounds that are likely to bind the HIV-1 CA hexamer - TRN-1 interface. This allowed the identification of a new family of inhibitors that specifically block HIV-1 nuclear import." It is not possible to understand what was done here and why the results are incongruent with what appears to be a different strategy.

Response: Indeed, some details were missing from both the M&M and Results sections and have now been included in the revised version of our manuscript, M&M (lines 824-872) and Results (lines 285-288) sections. This point was also discussed in detail above.

Additionally, the 3D-coordinates of the complex used for virtual screening can be freely accessible for download from [10.5281/zenodo.11060905](https://zenodo.org/record/11060905).

-Disassembly assay is not really a good representation of what is happening in the cellular context; The standard is live microscopy-for relevant assays Pathak, Melikian, Hope, and other labs.

Response: Our *in vitro* assays were not trying to infer the exact cellular function of H27, but merely to look at the effects of H27 and derivatives on the intrinsic properties of CA to both assemble and disassemble. We primarily conducted assembly assays as these are reasonably standard approaches and whilst they are not as common, we conducted disassembly assays to examine if we could detect any differential, properties between the compounds tested. We have not conducted microscopy studies, that we agree have been excellent from the labs referred to by the reviewer. Nevertheless, we do not believe it precludes us from extending our *in vitro* assays for capsid function from assembly to disassembly to examine for H27 effects.

-"Although we previously showed that in HIV-1 the glycine at position 89 is optimally positioned for binding to TRN-1 (Fernandez et al., 2019), G89V mutants were still sensitive to H27, confirming that H27 makes multiple contact points with the CA". The resistance experiments do not seem to confirm or be consistent with the *in silico* study.

Response: Actually, the resistance experiments, *in silico* study and NMR all agree that G89 is not a contact point for H27. We now mention this in the revised manuscript (line 312-313).

Referee #3 (Remarks for Author):

This manuscript describes the identification of H27 and structural analogues that were found to bind HIV-1 capsid (CA) and prevent import through the nuclear pore. The discovery of H27 is predicated upon the use of *in silico* approaches, but significant biological data is presented to validate the identification of this hit compound. An SAR study was also done using commercially available analogues of H27 to identify 5 additional active compounds that do not show significant cytotoxicity. This portion of the manuscript is somewhat limited as these compounds do not appear to be considered as viable leads for development since nearly all of the work was done with H27.

Response: It is true that the SAR did not identify viable leads. However, the SAR does provide some important insight into the Markush structure, and shows a remarkable specificity linked to the molecular structure of the compounds.

In addition, one limitation of this SAR work was in the limited variation of the "left" side of the molecules as shown in Figure EV6. The data obtained suggests that the right-hand of the molecule is not critical for activity, suggesting that key contacts are made in the other three rings. While the third ring is varied through the introduction of various substituents and a variety of substitution patterns, the first and second rings (from the left) are largely held constant, limiting the overall value of the conclusions that can be drawn. This is a relatively minor concern as this is not the focus of the paper.

Response: We agree that this is a limited SAR analysis. Only few compounds related to the initial hit were commercially available and acquired for the study (45 compounds). A better SAR analysis would have required custom synthesis of 100+ compounds but unfortunately, we did not have sufficient funds to initiate this program.

Overall, the paper is well constructed and identifies a molecule that acts via a novel mechanism. This could provide a nice starting point for development of a class of compounds that act at this particular site.

There are some minor issues that should be addressed, however, if this paper is accepted for publication:

Line 35-36: Add the word "as" in the phrase "as well as HIV-1 strains"

Response: This change has been made (line 32).

Line 37: Stating that this is a new class of antivirals suggests that these are approved therapeutics. It would be more appropriate to say that these compounds demonstrate a novel mechanism of action.

Response: This change has been made (lines 33-34).

Line 42: Recommend either "by active transport" or "by an active transport mechanism".

Response: This change has been made (line 39).

Line 129 (and after): It is recommended that bold font be used when indicating compound numbers.

Response: The text has been altered throughout accordingly.

Line 132: The molecular formula should be presented using subscripted numbers.

Response: This change has been made (line 130).

Table EV1: Compounds 8, 9, 17, and 24 are listed as NOT AVAILABLE. This would seem to indicate that those compounds could not be obtained for subsequent screening. If this is the case, why are they included in the list of compounds 1-40, which is stated to have been screened (Line 129).

Response: Yes, compounds #8, 9, 17 and 24 were not commercially available as powder or in solution and could not be evaluated experimentally, but they were present at the initial stage that is the *in silico* screening (thus as virtual compounds). It often happens that the content of chemical libraries used for docking is not updated as often as we would expect, or in some cases the stock runs out between two updates. The term "Not available" is replaced by "No longer in stock", in an amended **Table EV1 new**.

Line 223-226: It is not clear here what the authors are suggesting about the SAR here, particularly with respect to fluorination. Compounds 55, 49, and 84 are among the most active and show limited cytotoxicity. Compounds 80 and 81 are also fluorinated (in similar positions) and show significant cytotoxicity.

Response: We have edited the manuscript to state that "This preliminary SAR pinpoints important contribution to some chemical groups and their positioning, but more molecules will be needed to draw conclusions on the role of single substituents or specific substitution patterns in terms of activity and cytotoxicity" (lines 242-244).

Figure EV6: The order of the numbering of the compounds in part A is a bit confusing. If the authors wish to group compounds that share similar substitution patterns (e.g. an ortho, meta, and para series), consider renumbering so that these numbers are in sequential order. In the current arrangement, it becomes difficult to find certain numbers (e.g. compound 49, which follows 55 and 70 in the list). Also, the labels in part B should be reviewed. In particular, the "H also possible" is not clear since the arrow points to the methyl group of the methoxy. I believe that the "H" would refer to complete removal of that methoxy group, rather than simply the presence of a phenol. In addition, if using fluorine, chlorine should also be used. Alternatively, fluoride/chloride could be used, but since the reference is to the atom itself (rather than the molecule), fluorine/chlorine would likely be more appropriate. Finally, the meaning of "isopropyl of large sulfonamides" is not clear, specifically what is meant by a "large sulfonamide".

Response: We have changed the order of the numbering of the compounds in part A so that the lower numbers are at the top of the stacked lists, and compounds are listed in ascending order, to make it easier to find specific numbers. We have also edited the labels in part B as requested (**Fig. EV9 new**).

Line 227-228: The statement that the mechanism of action of the analogues is linked to their structure is the very definition of the term "structure-activity relationship". This statement, therefore, is a bit redundant and not necessary. The value of this sentence, however, is the recognition of the high degree of structural specificity required for activity.

Response: This statement has been edited as recommended (line 237).

15th Aug 2024

Dear Dr. Arhel,

Thank you for the submission of your revised manuscript to EMBO Molecular Medicine and please accept my apologies for the delay in getting back to you due to the holiday season. I am pleased to inform you that we will be able to accept your manuscript pending the following final amendments:

- 1) Please address the minor point raised by the referee. See below for the EV figure issue.
- 2) Authors: Please specify the corresponding author on the title page and provide the e-mail address.
- 3) Figures:
 - Please submit main and EV figures as high-resolution eps, .tif, or .jpg files and make sure that the figures are on one-page.
 - Please be aware that only 5 EV figure are allowed. Select 5 of current 16 EV figures to be presented as EV figures with their legends in the main manuscript file. Also, rename Supplementary Figure legends to Expanded View Figure Legends. 11 remaining EV figure should be compiled in the Appendix and renamed to Appendix Figure S1, etc. with their legends below the respective figure. Please also update their callouts in the main manuscript text. Appendix should have table of content on the first page and should be uploaded as a single PDF file.
- 4) Tables: Please rename Table EV1 to Dataset EV1 and the rest of EV tables to Table EV1-5. Remove their legends from the main manuscript file and place them in each excel file. The legend of the dataset can be provided as a separate sheet/tab. Please update their callouts in main manuscript.
- 5) Movies: Remove the legend from the manuscript and provide it as a readme.txt file (titled Movie EV1 instead of Movie S1). The movie file should be zipped together with the legend and uplidd as Movie EV1 zip folder.
- 6) In the main manuscript file, please do the following:
 - Please address all comments suggested by our data editors listed below:
 - o Figure legends:
 1. Please define the annotated p values ****/**** as well as provide the exact p-values for the same in the legend of figure 2g; EV 6a-b; as appropriate.
 2. Please note that the exact p values are not provided in the legends of figures 5d; 6a-b; EV 16d.
 3. Please indicate the statistical test used for data analysis in the legend of figure 2d.
 4. Please note that the box plot needs to be defined in terms of minima, maxima, centre, bounds of box and whiskers, and percentile in the legend of figure 2d.
 5. Please note that information related to n is missing in the legends of figures EV 14b; EV 16a-b.
 6. Please note that n=2 in figure EV 4d.
 7. Please note that the error bars are not defined in the legends of figures 4a-b; EV 4a; EV 14b; EV 16a-b.
 - The manuscript sections should be in the following order: Title page - Abstract & Keywords - Introduction - Results - Discussion - Methods - Data Availability - Acknowledgments - Disclosure Statement & Competing Interests - References - Figure Legends - (Main Tables with legends) - Expanded View Figure Legends.
 - Add up to 5 keywords.
 - Remove data not shown (p22).
 - Add "Disclosure and competing interests statement". We updated our journal's competing interests policy in January 2022 and request authors to consider both actual and perceived competing interests. Please review the policy <https://www.embopress.org/competing-interests> and update your competing interests if necessary.
 - Please include structured Methods section that includes a Reagents and Tools Table followed by a Methods and Protocols section. More information on how to adhere to this format as well as downloadable templates (.docx) for the Reagents and Tools Table can be found in our author guidelines: <https://www.embopress.org/page/journal/17574684/authorguide#structuredmethods> An example of a paper with Structured Methods can be found here: <https://www.embopress.org/doi/full/10.1038/s44320-024-00037-6#sec-4>
 - Rename Material and Methods to Methods.
 - In Methods, provide dilutions for each antibody used in the study.
 - In Methods, provide the statement that informed consent was obtained from all human subjects and confirm that the experiments conformed to the principles set out in the WMA Declaration of Helsinki and the Department of Health and Human Services Belmont Report. Please also include your response to Author Checklist raw 92-93.
 - In Methods, a statistical paragraph should reflect all information that you have filled in the Authors Checklist, especially regarding randomization, blinding, replication.
 - Indicate in legends number and nature of replicates and exact p= values, not a range, along with the statistical test used. To keep the figures "clear" some authors found providing an Appendix table Sx with all exact p-values preferable. You are welcome to do this if you want to.
 - Correct the reference citation in the reference list. Where there are more than 10 authors on a paper, 10 will be listed, followed by "et al.". Please check "Author Guidelines" for more information. <https://www.embopress.org/page/journal/17574684/authorguide#referencesformat>
- 7) Funding: Please make sure that information about all sources of funding are complete in both our submission system and in the manuscript. Cancer Research UK, UK Medical Research Council and Wellcome Trust: CC1078 is currently missing in our

submission system.

8) The Paper Explained: Please provide "The Paper Explained" and add it to the main manuscript text. Please check "Author Guidelines" for more information. <https://www.embopress.org/page/journal/17574684/authorguide#researcharticleguide>

9) Synopsis: Every published paper now includes a 'Synopsis' to further enhance discoverability. Synopses are displayed on the journal webpage and are freely accessible to all readers. They include separate synopsis image and synopsis text.

- Synopsis image: Please provide a visual abstract as a high-resolution jpeg file 550 px-wide x (250-400)-px high to illustrate your article.

- Synopsis text: Please provide a short standfirst (maximum of 300 characters, including space) as well as 2-5 one sentence bullet points that summarise the paper as a .doc file. Please write the bullet points to summarise the key NEW findings. They should be designed to be complementary to the abstract - i.e. not repeat the same text. We encourage inclusion of key acronyms and quantitative information (maximum of 30 words / bullet point). Please use the passive voice.

10) As part of the EMBO Publications transparent editorial process initiative (see our Editorial at <http://embomolmed.embopress.org/content/2/9/329>), EMBO Molecular Medicine will publish online a Review Process File (RPF) to accompany accepted manuscripts. This file will be published in conjunction with your paper and will include the anonymous referee reports, your point-by-point response and all pertinent correspondence relating to the manuscript. Let us know whether you agree with the publication of the RPF and as here, if you want to remove or not any figures from it prior to publication. Please note that the Authors checklist will be published at the end of the RPF.

11) Please provide a point-by-point letter INCLUDING my comments as well as the reviewer's reports and your detailed responses (as Word file).

I look forward to reading a new revised version of your manuscript as soon as possible.

Yours sincerely,

Zeljko Durdevic

*** Instructions to submit your revised manuscript ***

1) a .docx formatted version of the manuscript text (including Figure legends and tables)

2) Separate figure files*

3) supplemental information as Expanded View and/or Appendix. Please carefully check the authors guidelines for formatting Expanded view and Appendix figures and tables at <https://www.embopress.org/page/journal/17574684/authorguide#expandedview>

4) a letter INCLUDING the reviewer's reports and your detailed responses to their comments (as Word file).

5) The paper explained: EMBO Molecular Medicine articles are accompanied by a summary of the articles to emphasize the major findings in the paper and their medical implications for the non-specialist reader. Please provide a draft summary of your article highlighting

This may be edited to ensure that readers understand the significance and context of the research.

Please refer to any of our published articles for an example.

6) For more information: There is space at the end of each article to list relevant web links for further consultation by our readers. Could you identify some relevant ones and provide such information as well? Some examples are patient associations, relevant databases, OMIM/proteins/genes links, author's websites, etc...

7) Author contributions: the contribution of every author must be detailed in a separate section.

8) EMBO Molecular Medicine now requires a complete author checklist

(<https://www.embopress.org/page/journal/17574684/authorguide>) to be submitted with all revised manuscripts. Please use the checklist as guideline for the sort of information we need WITHIN the manuscript. The checklist should only be filled with page numbers where the information can be found. This is particularly important for animal reporting, antibody dilutions (missing) and exact values and n that should be indicated instead of a range.

9) Every published paper now includes a 'Synopsis' to further enhance discoverability. Synopses are displayed on the journal webpage and are freely accessible to all readers. They include a short stand first (maximum of 300 characters, including space) as well as 2-5 one sentence bullet points that summarise the paper. Please write the bullet points to summarise the key NEW findings. They should be designed to be complementary to the abstract - i.e. not repeat the same text. We encourage inclusion of key acronyms and quantitative information (maximum of 30 words / bullet point). Please use the passive voice. Please attach these in a separate file or send them by email, we will incorporate them accordingly.

You are also welcome to suggest a striking image or visual abstract to illustrate your article. If you do please provide a jpeg file 550 px-wide x 300-600px high.

10) A Conflict of Interest statement should be provided in the main text

11) Please note that we now mandate that all corresponding authors list an ORCID digital identifier. This takes <90 seconds to complete. We encourage all authors to supply an ORCID identifier, which will be linked to their name for unambiguous name identification.

Currently, our records indicate that the ORCID for your account is 0000-0001-5309-1725.

Link Not Available

12) Include a Reagents and Tools Table as part of the Methods section, which can be downloaded from our author guidelines (<https://www.embopress.org/page/journal/17574684/authorguide#structuredmethods>)

Photos 400-800 DPI

*Additional important information regarding figures and illustrations can be found at

<https://bit.ly/EMBOPressFigurePreparationGuideline>. See also figure legend preparation guidelines:

<https://www.embopress.org/page/journal/17574684/authorguide#figureformat>

***** Reviewer's comments *****

Referee #1 (Remarks for Author):

The authors have adequately responded to all of my queries, and I appreciate their detailed and thoughtful responses. In instances where additional data were not added, the authors effectively conveyed their reasoning. They have implemented the most important suggestions from the different reviewers, resulting in a significant improvement in the overall quality of the manuscript. I have only minor comments at this stage: several EV figures are not numbered, and there is a spelling mistake in line 139 of the manuscript ("overT Toxicity").

***** Reviewer's comments *****

Referee #1 (Remarks for Author):

The authors have adequately responded to all of my queries, and I appreciate their detailed and thoughtful responses. In instances where additional data were not added, the authors effectively conveyed their reasoning. They have implemented the most important suggestions from the different reviewers, resulting in a significant improvement in the overall quality of the manuscript. I have only minor comments at this stage: several EV figures are not numbered, and there is a spelling mistake in line 139 of the manuscript ("overT Toxicity").

The word "overt" was intended to mean "apparent" or "definite".

2nd Sep 2024

Dear Dr. Arhel,

We are pleased to inform you that your manuscript is accepted for publication and is now being sent to our publisher to be included in the next available issue of EMBO Molecular Medicine.
